# GraphPrivatizer: Improved Structural Differential Privacy for Graph Neural Networks

**Rucha Bhalchandra Joshi**[*]                                                            *rucha.joshi@niser.ac.in*
*National Institute of Science Education and Research, Bhubaneswar*
*Homi Bhabha National Institute, Mumbai*

**Patrick Indri**[*]                                                                  *patrick.indri@tuwien.ac.at*
*Research Unit Machine Learning*
*TU Wien, Vienna*

**Subhankar Mishra**                                                                    *smishra@niser.ac.in*
*National Institute of Science Education and Research, Bhubaneswar*
*Homi Bhabha National Institute, Mumbai*

**Reviewed on OpenReview:** *https://openreview.net/forum?id=lcPtUhoGYc*

## Abstract

Graph privacy is crucial in systems that present a graph structure where the confidentiality and privacy of participants play a significant role in the integrity of the system itself. For instance, it is necessary to ensure the integrity of banking systems and transaction networks, protecting the privacy of customers' financial information and transaction details. We propose a method called GraphPrivatizer that privatizes the structure of a graph and protects it under Differential Privacy. GraphPrivatizer performs a controlled perturbation of the graph structure by randomly replacing the neighbors of a node with other similar neighbors, according to some similarity metric. With regard to neighbor perturbation, we find that aggregating features to compute similarities and imposing a minimum similarity score between the original and the replaced nodes provides the best privacy-utility trade-off. We use our method to train a Graph Neural Network server-side without disclosing users' private information to the server. We conduct experiments on real-world graph datasets and empirically evaluate the privacy of our models against privacy attacks.

## 1 Introduction

In recent years, many research efforts have been made to effectively learn from graph-structured data. Graph-based approaches have been successful in a variety of tasks such as fake news detection in social networks (Benamira et al., 2019) and drug discovery (Gaudelet et al., 2021). Graphs can incorporate both information about individual data points and about their interactions: Graph Neural Networks (GNNs, Scarselli et al., 2008) have been in capturing this information and learning over graph-structured data. Both the information about the individual data points and the relational information can be, however, of a sensitive nature and must, therefore, be protected. Large scale machine learning models may require sending information to a server where the training is performed, which poses a privacy risk. Efforts have thus been recently made to address privacy attacks on graphs (Zhang et al., 2021; 2022). One possibility to protect private information on graphs is to use the formal privacy guarantees offered by Differential Privacy (DP, Dwork, 2006), whose range of applications on graph-structured data has been recently expanding (Mueller et al., 2022b). DP has been used both in centralized settings where a server has graph-wide access to information (Olatunji et al., 2023; Wu et al., 2022; Sajadmanesh et al., 2023) and in local settings (Sajadmanesh & Gatica-Perez, 2021).

---

[1]Equal contribution.

In a centralized privacy setting, a trusted entity is allowed to gather the private data and learn on it, while promising to release a model from which private information cannot be inferred. In this setting, the training procedure itself must therefore be DP (Abadi et al., 2016). A local privacy setting is instead desirable when no entity is trusted to gather all the private information: in this case, data must be privatized locally before it is made available to a central entity where the training is performed. The local privacy setting is therefore crucial in cases where no central entity is trusted to be willing and capable to keep private data secure. Once the data has been locally privatized, the central server is not able to infer private data and the training procedure itself does not need to be DP (Mueller et al., 2022b). Despite these advantages, a local privacy setting often results in reduced performance if compared to the centralized one (Cormode et al., 2018; Yang et al., 2023), due to the large amount of noise that the local privatization entails.

While recent efforts have been made towards improving the privacy-utility trade-off in a local privacy setting (Yang et al., 2023), little investigation on locally privatizing the edges of a graph which is then trained on a central entity (e.g., a server) has been previously performed. Motivated by real-world applications such as private learning on social network data, we therefore focus on protecting the relational information contained in graphs by means of local privatization techniques that act on the individual users' side.

To address this problem, we propose GraphPrivatizer, a method to locally privatize a graph's structure that does not rely on graph-wide information, and which protects the privacy of features and labels as well. In particular: (i) We introduce a definition of edge LDP for our local privacy settings; (ii) We propose GraphPrivatizer, a method that locally privatizes the structure of a graph while preserving the out-degree of nodes by construction and keeping labels and features private too; (iii) Taking advantage of the notion of message passing in GNNs, we parametrize the perturbations of a node's neighborhood which allows to only replace neighbors with other similar nodes to improve utility while preserving privacy; and (iv) We evaluate our proposal on different real-world datasets to investigate its privacy-utility trade-off, empirically assessing its privacy guarantees using privacy attacks that try to recover the private structure of the graph.

## 2 Related Work

GNNs have gained increasing popularity as the framework of choice to solve graph-based learning tasks in recent years. The efficacy of GNNs in graph representation learning has motivated the proposal of several GNN variants such as Graph Convolutional Networks (Zhang et al., 2019), Graph Attention Networks (Veličković et al., 2018), and GraphSAGE (Hamilton et al., 2017), as well as architectures for large multi-relation graphs (Iyer et al., 2021; Wang et al., 2019). Recent research efforts have also been made to address privacy attacks on graphs (Zhang et al., 2021; 2022). Privacy attacks can be categorized as graph properties attacks (inferring, e.g., the number of nodes), membership attacks (inferring, e.g., whether a subgraph is part of a graph), and graph reconstruction attacks (Zhang et al., 2022). Specifically, the existence of an edge between two nodes is often sensitive in nature (Mueller et al., 2022b) and should be kept private.

Differential Privacy (DP, Dwork, 2006) offers formal privacy guarantees to protect information about individual training points, and has been used to provide privacy guarantees in GNNs as well. DP has been utilized in centralized settings where a server has access to information on the entire graph (Olatunji et al., 2023; Wu et al., 2022; Sajadmanesh et al., 2023), and in local settings (Sajadmanesh & Gatica-Perez, 2021; Joshi & Mishra, 2022; 2023). Different formulations of DP on graphs aim at protecting the relationship between nodes (*edge*-level DP) (Raskhodnikova & Smith, 2016; Hidano & Murakami, 2022), the individual nodes themselves (*node*-level DP) (Raskhodnikova & Smith, 2016; Ayle et al., 2022; Kasiviswanathan et al., 2013; Olatunji et al., 2023), or the entire graph as a single entity (*graph*-level DP) (Mueller et al., 2022a). For a survey on recent advances in DP approaches on structured data, refer to Mueller et al. (2022b).

In this work, we focus on protecting the structure of the graph, i.e., hiding its edges. Previous work has addressed structural privacy using central (Sajadmanesh et al., 2023; Olatunji et al., 2023) or local (Joshi & Mishra, 2022; 2023; Hidano & Murakami, 2022) DP; these approaches require however some entity to have access to the entire noiseless adjacency matrix of the graph (Sajadmanesh et al., 2023; Joshi & Mishra, 2022) or to part of it (Joshi & Mishra, 2023; Hidano & Murakami, 2022) in order to privatize the graph structure. Such approaches may thus pose privacy concerns or depend on the availability of public data (Olatunji et al., 2023). Additionally, approaches such as Sajadmanesh et al. (2023) require the introduction of a custom

architecture. Hidano & Murakami (2022) propose a degree-preserving randomized response algorithm for graph classification on unattributed graphs where they empirically show the benefit of preserving the degree of nodes after graph perturbations. Besides the different learning task they consider, as we will focus on node classification for graphs with node features and labels, Hidano & Murakami (2022) require nonetheless that each node is aware of how many nodes the graph contains. Joshi & Mishra (2023) is, to the best of our knowledge, the closest existing approach that guarantees edge-level local DP: their approach requires however that each node has noiseless access to portions of the adjacency matrix and comes at a substantial reduction in model performance if compared to a non-private model. Additionally, Joshi & Mishra (2023) do not test against privacy attacks that try to recover the edges of the graph. In comparison, our approach can be used with any conventional GNN architecture and we do not rely on public data: we adopt a local privacy setting where the individual nodes have noise-free access only to their own edges, features, and labels, and where no entity has access to the complete adjacency matrix of the graph.

## 3 Preliminaries and Problem Statement

In this section we recall the definitions of Graph Neural Network (GNN, Scarselli et al., 2008) and Local Differential Privacy (LDP, Dwork, 2006; Yang et al., 2020). Then, we briefly discuss randomized response (RR, Warner, 1965) and edge privacy in graphs, as well as LinkTeller (Wu et al., 2022) as the privacy attack we use to validate our approach. Finally, we describe our local privacy setting and problem statement.

### 3.1 Graph Neural Networks

Consider an unweighted graph defined as a tuple $G = (V, E, X, Y)$, where $V = V_L \cup V_U$ is the set union of labeled nodes $V_L$ and unlabelled nodes $V_U$, $E$ is the set of edges, $X \in R^{|V| \times d}$ is a feature matrix consisting of $d$-dimensional feature vectors, one for each node $v \in V$, and $Y$ is the set of labels. Let $\mathcal{N}(v)$ denote the *neighborhood* of $v$, that is, the set of nodes which are adjacent to $v$. Let $\deg(v)$ denote the *degree* of $v$, that is, the size of its neighborhood. Graph Neural Networks (GNNs, Scarselli et al., 2008) are a class of models that have been effective in learning over graph-structured data. A typical GNN consists of $L$ layers where the embeddings of the nodes in a certain layer are obtained from the previous layer by means of an *aggregation* and an *update* function. Specifically, the embedding $h_v^l$ for a node $v$ in layer $l$ is obtained by aggregating the embeddings of its neighbors $\mathcal{N}(v)$ from layer $l-1$ and passing the resulting aggregated message $m_v^l$ through the update function. The aggregation function is a permutation invariant and differentiable function, while the update function is a trainable and non-linear function:

$$m_v^l = \text{AGGREGATE}(\{h_u^{l-1} \mid \forall u \in \mathcal{N}(v)\}), \tag{1}$$

$$h_v^l = \text{UPDATE}(\{h_v^{l-1}, m_v^l\}). \tag{2}$$

Common choices for the AGGREGATE function are the sum or the mean, while the UPDATE function can be, for instance, a neural network.

### 3.2 Differential Privacy

Differential Privacy (DP, Dwork, 2006) is a formal definition of privacy that protects individual training points. As originally introduced by Dwork (2006), *central* or *global* DP, simply referred to as DP, was designed for a *centralized* setting where a trusted entity gathers all user data and guarantees to process it and produce an output while preserving the privacy of users. More formally, DP guarantees that an attacker cannot confidently infer whether the output of a DP mechanism $\mathcal{M}$ was obtained from a database $D$ or from a database $D'$, where $D$ and $D'$ differ in a single record and are thus said to be *adjacent* datasets. In a *local* privacy setting no trusted entity can process the private data of users. In this more restrictive case each user perturbs its data locally and provides the central entity with a noisy version of its data only. In a local privacy setting the datasets $D$ thus consist of data from individual users which is then protected under Local Differential Privacy (LDP, Yang et al., 2020).

**Definition 3.1** ($\epsilon$ LDP)**.** *Let $\epsilon > 0$. Consider a randomized mechanism $\mathcal{M} : \mathcal{D} \to \mathcal{R}$ and probabilities* Pr *taken over the coin tosses of $\mathcal{M}$. $\mathcal{M}$ satisfies $\epsilon$ local differential privacy if, for any possible pairs of user's private data points $x$, $x' \in \mathcal{D}$ and for any possible outputs $S \subseteq \mathcal{R}$:*

$$\Pr[\mathcal{M}(x) \in S] \leq e^\epsilon \Pr[\mathcal{M}(x') \in S].$$

We refer to $\epsilon$ as the *privacy budget* of the algorithm. In particular, $\epsilon = 0$ indicates that the randomized mechanism is perfectly private and implies that the output of the mechanism is independent of the input. On the other hand, $\epsilon = \infty$ provides no privacy guarantee. The choice of $\epsilon$ is both problem and data dependent (Lee & Clifton, 2011), with common ranges often considering values $\epsilon \in (0, 10]$ (Wu et al., 2022; Sajadmanesh & Gatica-Perez, 2021). For a given deterministic function, DP can be achieved by adding random noise to the output of the function to hide the contribution of individual training points, where the amount of noise added depends on the choice of privacy budget $\epsilon$ (Dwork et al., 2014). As no single entity is trusted with all the users' private data, LDP is generally a stronger privacy model than DP. However, the lack of such trusted central entity and thus the need to add noise to the local data of each user before communicating it to the central entity entails a greater total noise which can negatively affect model performance (Cormode et al., 2018). This, in turn, enhances the importance of novel approaches that can provide LDP with good accuracy which are thus our focus.

### 3.3 DP and privacy attacks in GNNs

DP can be applied to graph context by defining a notion of adjacency for graphs. Considering the (centralized) DP setting first, we say two graphs $G$ and $G'$ are *edge adjacent* if they differ exactly in one edge, that is, if $G'$ can be obtained from $G$ by adding or removing a single edge. Similarly, $G$ and $G'$ are *node adjacent* if they differ in exactly one node, that is, if $G'$ can be obtained from $G$ by adding or removing a single node and its edges. In our local privacy setting we consider LDP which can guarantee privacy on graphs in the sense of Definition 3.1 on any pair of adjacent user inputs. Focusing here on edge privacy, one can define a notion of edge adjacency to protect a user's edges. A common definition of edge LDP considers a user's $v$ neighbor list as represented by $|V|$-dimensional bit vector $(b_1, \ldots, b_{|V|})$, where $b_{i,i=1,\cdots|V|} = 1$ if and only if there is an edge between node $v$ and node $v_i$, and $b_i = 0$ otherwise.

**Definition 3.2** ($\epsilon$-edge LDP, (Qin et al., 2017))**.** *Let $\epsilon > 0$. A randomized mechanism $\mathcal{M} : \mathcal{D} \to \mathcal{R}$ satisfies $\epsilon$-edge local differential privacy if, for any possible pairs of user's neighbor lists $b$, $b'$ differing by one bit, and for any possible outputs $S \subseteq \mathcal{R}$, it holds that:*

$$\Pr[\mathcal{M}(b) \in S] \leq e^\epsilon \Pr[\mathcal{M}(b') \in S]$$

Edge LDP can be achieved by perturbing the neighbor list of each node using *randomized response* (RR, Warner, 1965; Qin et al., 2017). RR flips every bit of each neighbor list with probability $1/(e^\epsilon + 1)$, guaranteeing $\epsilon$-edge LDP. However, as large values of $\epsilon$ are undesirable because they imply low privacy, and as the neighbor list is often sparse, RR increases the connectivity of the graph. The addition of many spurious edges has negative consequences on the performance of a GNN trained on the perturbed graph (Joshi & Mishra, 2022): thus, we develop different perturbation techniques which offer a better trade-off between privacy and model accuracy for graph data. In this work, we propose an alternative definition of adjacency for edge privacy which imposes additional conditions on the edges RR can act on.

To validate our approach, we test it against a LinkTeller attack. LinkTeller (Wu et al., 2022) is an influence analysis based attack that recovers private edges from trained GNNs. The attack assumes that the trained GNN exposes an inference API; at test time, a user can provide node features to query the API and obtain predictions for said nodes. For each pair of nodes the attacker provides perturbed feature vectors for the first node and queries the API with them, evaluating the effect this has on the predictions of the second node. The attacker then guesses the presence of edges between the pairs of nodes which have a high influence on each other. See Wu et al. (2022) for more details on LinkTeller. As LinkTeller cannot deal with randomized models such as GraphSAGE (Wu et al., 2022), we instead attack it using the LSA2 attack described in He et al. (2021). LSA2 leverages on node-level information and computes the model *posterior* for every node the attacker possesses, assigning edges to pairs of nodes whose posteriors have a high correlation.

### 3.4 Problem statement

We address private learning on graphs with LDP where each node can is considered as an individual user: for a graph $G = (V, E, X, Y)$ we consider $V$, $E$, $X$, and $Y$ to be private to individual nodes/users, which only share noisy versions of them with a server. The server uses the information to train a GNN for node classification, learning to predict labels $Y$ in a semi-supervised learning setting. More explicitly, no entity (neither users/nodes, nor the server) has complete and noise-free information about the graph. The only noiseless information a node can have access to is its own features, label, and edges (Figure 1). We elect this as our setting of choice because of its similarities with real-world scenarios where individual nodes/users may not have knowledge about the entire graph beyond the nodes/users they directly interact with.

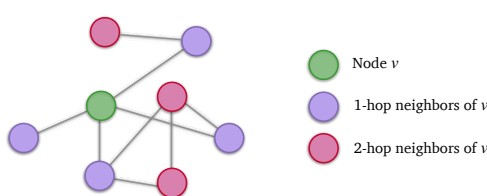

Figure 1: In our setting, a node $v$ knows only about its immediate, 1-hop neighbors. In fact, a node may not be aware of its 2-hop neighbors in a real-world scenario.

## 4 Approach

In this section, we introduce our approach, called GraphPrivatizer, which provides edge, feature, and label privacy. Specifically, we focus on investigating the trade-off between edge privacy and GNN performance in the local privacy setting described in Section 3.4. A global notion of differential privacy is not allowed by our local privacy setting, as no central entity is trusted with the entire graph. $\epsilon$-edge LDP (Definition 3.2) too has limitations in our privacy setting, as we will discuss in the next section. Therefore, we provide a new definition of edge privacy for our setting and propose a novel edge private algorithm based on it. To guarantee label and/or feature privacy, GraphPrivatizer uses existing techniques presented in Section 4.3.

### 4.1 Adjacent neighborhoods

The definition of $\epsilon$-edge LDP provided in Section 3.3 entails the perturbation of the neighbor list of a node $v$ using RR, where two neighbor lists are adjacent if they differ by one bit. As previously mentioned, this perturbation leads to a great increase in the connectivity of the graph for small, thus desirable, values of $\epsilon$. Moreover, in the local privacy setting described in Section 3.4 a node's neighbor list include only its immediate neighbors and there are therefore no graph-wide neighbor lists to perturb with the standard RR approach. To define a notion of privacy which is appropriate for our setting we choose to consider the *set* of neighbors of node $v$: two neighbor sets are said to be adjacent if they differ by a single node.

**Definition 4.1** (Adjacent neighborhoods). *Consider a node $v$. Let $b = \{v_1, \ldots, v_d\} = \mathcal{N}(v)$, $b' = \{v'_1, \ldots, v'_d\} = \mathcal{N}'(v)$ be two neighbor sets, with $d = \deg(v)$. We say $\mathcal{N}(v)$ and $\mathcal{N}'(v)$ are adjacent if they differ in only one element; that is, they are adjacent neighborhoods if, without loss of generality, $v_1 \neq v'_1$ and $v_i = v'_i$ for $i = 2, \ldots, d$.*

We use Definition 4.1 to introduce an *edge set* notion of LDP, which we refer to as $\epsilon$-edge set LDP.

**Definition 4.2** ($\epsilon$-edge set LDP). *Let $\epsilon > 0$. A randomized mechanism $\mathcal{M} : \mathcal{D} \to \mathcal{R}$ satisfies $\epsilon$-edge set local differential privacy if, for any possible pairs user's neighbor sets $b$, $b'$ that are adjacent according to Definition 4.1, and for any possible outputs $S \subseteq \mathcal{R}$, it holds that:*

$$\Pr[\mathcal{M}(b) \in S] \leq e^\epsilon \Pr[\mathcal{M}(b') \in S]$$

$\epsilon$-edge set LDP can be seen as a relaxation of $\epsilon$-edge LDP Definition 3.2, as it practically entails a more controlled perturbation of the edges of a node. To be more explicit on the relation between $\epsilon$-edge set LDP (Definition 4.2) and the $\epsilon$-edge LDP (Definition 3.2) notion used by related work such as Joshi & Mishra (2023), note that one can obtain any set perturbation as described in Definition 4.1 by means of two bit flips on the neighbor list of a node. A $2\epsilon$-edge LDP mechanism is thus also $\epsilon$-edge set LDP. The converse does not hold as not all two bit flips on the neighbor list of a node correspond to a set perturbation as

described in Definition 4.1. Moreover, and in contrast with $\epsilon$-edge LDP, adjacent neighbor sets according to Definition 4.1 have the same number of nodes. Thus, a perturbation of the neighborhoods based on this notion of adjacency preserves, by construction, the degree of nodes. The preservation of node degrees after private perturbations is a desirable property as empirically shown, albeit for the different task of graph classification on unattributed graphs, by (Hidano & Murakami, 2022).

### 4.2 Edge privacy

Equipped with the notion of adjacency in Definition 4.1, we develop a perturbation technique that acts on the neighbor set of a node which is edge set LDP in the sense of Definition 4.2. Our approach informally seeks to replace nodes in a neighbor set with other nodes that are *similar* with respect to some similarity measure. The perturbed neighborhood can in this way retain more of the original information content of the neighborhood and provide good performance. Specifically, our proposal randomly replaces a neighbor $u$ of a node $v$ with one of the neighbors of $u$ itself. That is, we perform perturbations considering nodes in the two-hop extended local view (Sun et al., 2019) of $v$. Two neighborhoods of $v$ are then adjacent according to Definition 4.1 if one can be obtained from the other by means of an edge perturbation within the two-hop expended local view of $v$ which preserves the degree of $v$. The replacement itself then occurs via RR, which ensures the privacy of the procedure. Conceptually, we can describe our method as consisting of two steps. The neighbor set $\mathcal{N}(v)$ of a node $v$ is perturbed by: *(i)* selecting a set of candidate replacement nodes for the neighbors of $v$ and *(ii)* randomly picking of the replacement nodes using RR. Algorithm 1 describes the procedure. We provide a summary of the main notation for ease of read.

| **Algorithm 1** Perturb neighborhood |
| --- |
| **Input:** Graph $G = (V, E, X, Y)$, node $v \in V$, similarity $s_\alpha$, threshold $\delta$, aggregation coeff. $\alpha$, strategy $g(s_\alpha, \alpha, \delta)$ |
| **Output:** $\mathcal{N}'(v)$: Perturbed neighborhood of node $v$ |
| 1: $\mathcal{N}(v) \leftarrow \text{GetNeighbors}(v, 1)$ |
| 2: $\mathcal{N}'(v) \leftarrow \varnothing$ |
| 3: **for** $u \in \mathcal{N}(v)$ **do** |
| 4: $\quad u' = \text{RR}(u, \text{QuerySimilar}(G, u, s_\alpha, \alpha, \delta, g))$ |
| 5: $\quad \mathcal{N}'(v) \leftarrow \mathcal{N}'(v) \cup u'$ |
| 6: **end for** |
| 7: **return** $\mathcal{N}'(v)$ |

| **Notation** | |
| --- | --- |
| $\delta$ | threshold on similarity |
| $\alpha$ | aggregation coefficient |
| $x_u$ | feature vector of node $u$ |
| $x_{u,\alpha}$ | aggregated feature vector of node $u$ and $\mathcal{N}(u)$, eq. (3) |
| $s_\alpha(v, u)$ | cosine similarity between $x_{v,\alpha}$ and $x_{u,\alpha}$, eq. (4) |

Consider a node $v$ and assume we want to perturb its neighbor set $\mathcal{N}(v)$ by replacing some of its nodes. Nodes $u \in \mathcal{N}(v)$ are randomly replaced with nodes picked in a set of candidates, where the candidates are selected from nodes in the two-hop extended local view of $v$ according to a similarity measure $s$. That is, the set of nodes which are considered as candidates to replace a node $u$ is constituted of nodes $u'$ that are similar to $u$. Given our local privacy setting, non-parametric and non-learnable similarity measures are a natural choice as no prior information on the data is available. We provide a comparison of the performance of our approach using the Euclidean distance and the cosine similarity in Appendix B, and find that the cosine similarity is preferable.

We therefore measure the similarity between $u$ and a candidate $u'$ using the cosine similarity $s$ of their feature vectors $x_u$ and $x_{u'}$, $s = \frac{x_u \cdot x_{u'}}{\|x_u\| \|x_{u'}\|}$. In this regard, we devise two strategies to obtain the set of candidates based on similarity, which are described in Algorithm 3 and Algorithm 4. For both strategies, only the nodes $u'$ which have a similarity score exceeding a threshold $\delta$ are selected as the set of candidates. As GNNs perform aggregations of the features of neighbors produce embeddings, we additionally propose to use such aggregated features to compute similarity scores. We therefore evaluate the similarity of a node $u$ using $\text{AGG}_u = \text{AGGREGATE}(\{x_n : \forall n \in \mathcal{N}(u)\})$ instead of $x_u$. We denote with $\alpha$ the hyper-parameter which parameterizes the contribution of aggregated features in the similarity computation. That is, for each node $u$ we compute the aggregated feature vector $x_{u,\alpha}$ of node $u$ and $\mathcal{N}(u)$ as

$$x_{u,\alpha} = (1 - \alpha)x_u + \alpha\text{AGG}_u. \tag{3}$$

The similarity between two nodes $u$ and $u'$ is then computed as

$$s_\alpha(u, u') = \frac{x_{u,\alpha} \cdot x_{u',\alpha}}{\|x_{u,\alpha}\|\|x_{u',\alpha}\|}. \tag{4}$$

To summarise, $\delta > 0$ can be used to filter out dissimilar replacement candidates, while $\alpha > 0$ is used to introduce aggregate information in the similarity computation. The case $\alpha = \delta = 0$ introduces no thresholds for the application of RR and no aggregation, and will thus be used as our reference to investigate the impact of such thresholds and aggregations. When computing similarities/aggregations between nodes, *noisy* feature vectors obtained according to Section 4.3 are used, ensuring that feature privacy is not violated. Our method is thus consistent with the setting described in Section 3.4 as we assume that the only noiseless information a node has access to consists of its own features, labels, and edges to 1-hop neighbors. Algorithm 2 describes the procedure to select replacement candidates.

---

**Algorithm 2** QuerySimilar

---

**Input**: Graph $G = (V, E, X, Y)$, node $v \in V$, similarity $s_\alpha$, threshold $\delta$, aggregation coeff. $\alpha$, strategy $g(s_\alpha, \alpha, \delta)$
**Output**: $\mathcal{S}(v)$: set of nodes similar to $v$, according to $s_\alpha$
 1: $\mathcal{S}(v) \leftarrow \varnothing$
 2: $\mathcal{S}(v) = g(v, s_\alpha, \alpha, \delta)$      # get similar nodes according to strategy $g$
 3: **return** $\mathcal{S}(v)$

---

As anticipated, we utilize two different strategies $g$ to select candidates for a node $u$ based on their similarity. Specifically, we either consider $u$'s most similar neighbor if it exceeds the similarity threshold $\delta$ as a candidate for replacement (Algorithm 3) or all the neighbors that exceed the similarity threshold $\delta$ (Algorithm 4).

---

**Algorithm 3** Most-similar neighbor

---

**Input**: node $v$, its neighborhood $\mathcal{N}(v)$, similarity function $s_\alpha$, threshold $\delta$, aggregation coeff. $\alpha$
**Output**: $m_v$: $v$'s most similar neighbor

 1: $m_v = \underset{u \in \mathcal{N}(v)}{\arg\max}\, s_\alpha(v, u)$
 2: **if** $s_\alpha(v, m_v) \geq \delta$ **then**
 3:     **return** $m_v$
 4: **end if**
 5: **return** $v$

---

**Algorithm 4** Threshold-based similar neighbors

---

**Input**: node $v$, its neighborhood $\mathcal{N}(v)$, similarity function $s_\alpha$, threshold $\delta$, aggregation coeff. $\alpha$
**Output**: $\mathcal{T}(v)$: set of neighbors similar to $v$
 1: $\mathcal{T}(v) \leftarrow \varnothing$
 2: **for** $u \in \mathcal{N}(v)$ **do**
 3:     **if** $s_\alpha(v, u) \geq \delta$ **then**
 4:         $\mathcal{T}(v) \leftarrow \mathcal{T}(v) \cup u$
 5:     **end if**
 6: **end for**
 7: **return** $\mathcal{T}(v)$

---

For each node $v$, the number of similarity values which need to be computed to perturb its neighborhood depends on the number of nodes in the two-hop extended local view of $v$ and is upper bounded $\sum_{u \in \mathcal{N}(v)} deg(u)$ which may thus be computationally expensive for very dense graphs. However, in a practical setting where nodes are distributed among different computing units, the neighbor perturbation is performed locally by the individual nodes. Moreover, the nodes only require the (perturbed, possibly aggregated) features of their neighbor to compute their similarity with them. In terms of communication cost, this amounts to two applications of the AGGREGATE (Section 3.1) function and is done as a pre-processing step before training. With these considerations and the observation that real-world datasets have a small average degree (Table 4), we expect an efficient implementation of our approach to scale well to sparse large graphs.

Once a set of candidate nodes has been obtained, the neighbor perturbation is performed in Algorithm 1 with RR. The probabilities of replacement associated with RR differ whether we consider one candidate replacement, (strategy in Algorithm 3) or a set of candidate replacements (strategy in Algorithm 4). Denote with $\Pr[u \to u']$ the probability that node $u$ gets replaced with node $u'$. If we consider the most similar replacement candidate only, RR is applied as follows.

$$\Pr[u \to u'] = \begin{cases} \frac{e^\epsilon}{e^\epsilon + 1} & \text{if } u' = u \\ \frac{1}{e^\epsilon + 1} & \text{otherwise} \end{cases} \tag{5}$$

If we consider the threshold-based strategy, RR is instead applied as follows.

$$\Pr[u \to u'] = \begin{cases} \frac{e^\epsilon}{e^\epsilon + d - 1} & \text{if } u' = u \\ \frac{1}{e^\epsilon + d - 1} & \text{otherwise} \end{cases} \tag{6}$$

To distinguish the two strategies, we refer to our method as GraphPrivatizer-m (GP-m) when using the most-similar strategy, and as GraphPrivatizer-t (GP-t) when using the threshold-based strategy. Regardless of the strategy, our approach is $\epsilon$-edge set private.

**Theorem 4.3.** *Algorithm 1 is $\epsilon$-edge set LDP.*

*Proof.* Algorithm 1 with the neighbor selection described in Algorithm 3 is a randomized mechanism based on RR: we denote it as $\mathcal{M}$. The proof follows from the DP of RR (see, e.g., Qin et al., 2017). Denote with $\Pr[v \to u]$ the probability that a node $v$ gets replaced by a node $u$, let $p = \frac{1}{e^\epsilon + 1}$ be the probability of node replacement according to RR, and $q = 1 - p$. Note that $\epsilon > 0$ implies $q > p$. Let $b = \{v_1, \ldots, v_d\}$, $b' = \{v'_1, \ldots, v'_d\}$ be two neighbor sets which differ in only one element; assume, without loss of generality, that $v_1 \neq v'_1$. Then, given any output $s = \{s_1, \ldots, s_d\}$ of $\mathcal{M}$, it holds that:

$$\frac{\Pr[\mathcal{M}(b) = s]}{\Pr[\mathcal{M}(b') = s]} = \frac{\Pr[v_1 \to s_1] \cdots \Pr[v_n \to s_d]}{\Pr[v'_1 \to s_1] \cdots \Pr[v'_n \to s_d]} = \frac{\Pr[v_1 \to s_1]}{\Pr[v'_1 \to s_1]} < \frac{q}{p} = e^\epsilon$$

With analogous reasoning (see, e.g., Wang et al., 2016), one can show that Algorithm 1 with the neighbor selection described in Algorithm 4 is $\epsilon$-edge set private. $\square$

## 4.3 Feature and label privacy

In addition to edge privacy, GraphPrivatizer also ensures feature and label LDP. That is, for each node, GraphPrivatizer ensures that an attacker cannot confidently infer the feature vector or the label. Specifically, we make use of the Drop algorithm introduced in Sajadmanesh & Gatica-Perez (2021), which enables efficient LDP GNN training with both private labels and node features. Features are privatized with a *multi-bit mechanism*, which allows individual nodes to perturb their features before communicating them. Labels are, instead, privatized using RR: a node's class is randomly replaced with one of the other available classes with the same approach described in Equation (6). We refer the reader to the original publication for more details. We assign a privacy budget $\epsilon_x$ for feature privacy and $\epsilon_y$ for label privacy.

## 4.4 Complete architecture

Figure 2 summarizes the complete approach: GraphPrivatizer produces perturbed features $X'$, edges $E'$, and labels $Y'$, which are then shared with a server to train a GNN. In particular, Figure 2 schematically shows which components of GraphPrivatizer have access to the unperturbed, private data.

**Theorem 4.4.** *GraphPrivatizer is $\epsilon + \epsilon_x + \epsilon_y$ LDP*

*Proof.* The private feature vectors are only used by the multi-bit mechanism, the private labels are only used by the RR mechanism for labels, and the private edge information is only used by Algorithm 1. In particular, Algorithm 1 only post-processes the privatized feature vectors and, due to the composition and robustness to post-processing properties of DP (Dwork et al., 2014), GraphPrivatizer is thus $\epsilon + \epsilon_x + \epsilon_y$ LDP. $\square$

## 5 Experiments

In this section, we empirically investigate[1] the performance of GraphPrivatizer and assess the trade-off between edge privacy and GNN accuracy in node classification tasks across several datasets. In particular, we (i) compare our approach against the baseline described in Section 5.1, and (ii) analyze the effects of the

---

[1]Code available at github.com/pindri/gnn-structural-privacy.

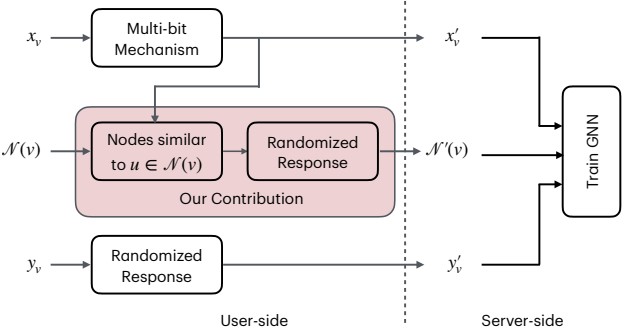

Figure 2: Scheme of GraphPrivatizer and how it acts on the features, edges, and labels of a node $v$. The shaded area highlights our contribution and where the algorithms we propose are utilized.

parameters $\alpha$ and $\delta$ on the privacy-accuracy trade-off for GraphPrivatizer. With respect to this trade-off, as discussed in Section 4.2, we use the setting $\alpha = \delta = 0$ as our reference and perform comparisons with $\alpha, \delta > 0$ to evaluate the benefits of higher thresholds $\delta$ and aggregation $\alpha$ coefficients. We empirically evaluate the edge privacy of our approach against attacks that try to recover the private edges. We use a variety of GNN architectures that include traditional convolutional GNNs, graph attention networks, and transformer networks, and perform experiments on the most commonly used benchmark datasets for node classification that include citation, co-purchase, and social networks. We experiment with GCN (Kipf & Welling, 2017), GraphSAGE (Hamilton et al., 2017), GAT (Veličković et al., 2018), GT (a graph transformer adapted from Shi et al. (2021)), GATv2 (Brody et al., 2022) and GraphConv (the graph convolution operator introduced in Morris et al. (2019)) on the Cora (Yang et al., 2016), Pubmed (Yang et al., 2016), LastFM (Rozemberczki & Sarkar, 2020), Facebook (Rozemberczki et al., 2021), and Amazon Photo (Shchur et al., 2018) datasets. In what follows, we will take the feature privacy budget $\epsilon_x$ and label privacy budgets $\epsilon_y$ to be fixed: the results and discussion will therefore focus on the edge privacy parameter $\epsilon$ which is simply referred to as privacy budget. We leave additional details on hyper-parameters and on the datasets used to Appendix A.

With regard to the privacy attack, we assume the trained GNN exposes an inference API that an attacker can query. We assume the attacker possesses feature information on pairs of nodes and wishes to determine whether an edge connects each pair of nodes. It should be noted that, according to the data model we adopt and describe in Section 3.4, no entity, either users or server, possesses noise-free information about other nodes' features. For this reason, we assume that the attacker itself may only have access to the noisy feature vector that a node sends to the server for, e.g., training. With these assumptions, we attack all models with LinkTeller (Wu et al., 2022) except for the GraphSAGE, which we attack with LSA2 (He et al., 2021; Wu et al., 2022). For LinkTeller, we use the default influence and graph density parameters of 0.001 and 1 (Wu et al., 2022). In all cases, we randomly sample 500 pairs of nodes that are connected in the original, unperturbed graph, as well as 500 pairs of nodes that are not connected in the original, unperturbed graph. The task of the attacker is to decide which of these nodes are connected or not connected, in a binary classification problem. We evaluate the performance of the attacker using the AUC, which we report multiplied by a factor $10^2$, that is, AUC $\in [0, 100]$. LinkTeller is a threshold-based binary classifier, so we use the AUC as a performance measure to capture all threshold values. A higher AUC denotes a higher ability of the attack to correctly identify the edges of the graph, and thus lower edge privacy, where an increase of 1 AUC point can be interpreted as a 1% increase in the likelihood of correctly identifying edges. It should be noted that the attacker has access to the API of a GNN which was trained on *perturbed* data, while we evaluate the attack AUC with respect to the original *unperturbed* data, which is what should be protected.

## 5.1 Comparison against baseline approaches

We compare GraphPrivatizer against a baseline approach adapted from existing literature on private node classification. Specifically the baseline approach is adapted from Sajadmanesh & Gatica-Perez (2021) and Wu et al. (2022) to our setting. Specifically, for the baseline we consider a slightly relaxation of the setting described in Figure 1 and assume that each node has access to the list of nodes in its 2-hop neighborhood. We

then apply randomized response with privacy budget $\epsilon$ to this list, thus perturbing the 2-hop neighborhood of the node. This approach corresponds to a local version of the *EdgeRand* approach described in Wu et al. (2022) and applied to each node. These adaptations are necessary as the original algorithms assume access to the full adjacency matrix which is not compatible with our local privacy setting. A very similar approach is also used in Joshi & Mishra (2023), who apply RR to 2-hop neighborhoods as well. We refer to this baseline simply as a *randomized response* baseline and denote it with RR. The baseline approach does not, in general, preserve the degree of the nodes and thus the sparsity of the adjacency matrix of the graph. Additionally, no form of threshold or aggregation is considered for the baseline. We highlight that the baseline approach assumes that nodes have access to the list of nodes in the entire 2-hop neighborhood, while GraphPrivatizer operates with the more strict (and private) condition that only immediate neighbors are known. Our experiments show that, despite this more strict setting, GraphPrivatizer outperforms the baseline in terms of accuracy with comparable privacy and thus offers a better privacy-utility trade-off. We additionally compare our results with LPGNN (Sajadmanesh & Gatica-Perez, 2021), which is mirrored by our experiments with $\epsilon = \infty$ where no edge privacy is guaranteed. For all comparisons we set the threshold and aggregation parameters for GraphPrivatizer to $\delta = 0$ and $\alpha = 0.5$ respectively. With reference to the results in Section 5.2, $\delta = 0$ ensures the best privacy as it imposes no threshold during the neighborhood perturbation, while $\alpha = 0.5$ corresponds to an intermediate amount of aggregation which provides good accuracy without sacrificing privacy. Refer to Section 5.2 for more details on the effects of $\alpha$ and $\delta$.

We present the results of our comparison between GraphPrivatizer (GP) and the RR baseline in Table 1: our approach is better in terms of accuracy without sacrificing privacy across all datasets. We performed statistical testing using a paired Wilcoxon signed rank test to compare the accuracy and privacy of our approach against the baseline across all privacy budgets, and report the p-values $P$ in Table 1a. For accuracy, we tested the null hypothesis $H_0$: $\text{Acc}_{\text{GP}} - \text{Acc}_{\text{RR}} = 0$ and found that GraphPrivatizer has higher accuracy across all datasets with statistical significance. For attack performance, we tested the null hypothesis $H_0$: $\text{AUC}_{\text{GP}} - \text{AUC}_{\text{RR}} = 0$ and found that there is no statistically significant difference in the privacy of our approach and the RR baseline across all datasets. In Table 1b we use GAP to denote the utility gap, i.e., the accuracy loss with respect to the non-edge-private, $\epsilon = \infty$ setting corresponding to LPGNN (Sajadmanesh & Gatica-Perez, 2021). While the non-edge-private setting is expected to provide better accuracy than the edge-private one, we are interested in evaluating how closely GraphPrivatizer and RR can match its accuracy while providing edge privacy. We report results for $\epsilon = 0.1$ and find that GraphPrivatizer narrows the utility gap when compared to the RR baseline across all datasets.

Table 1: Aggregate results across models for GraphPrivatizer (GP) and a randomized response (RR) baseline. For GraphPrivatizer, we report results for $\alpha = 0.5$ and $\delta = 0$. Sub-table (a) reports the average improvement in accuracy of GraphPrivatizer over the baseline as well as the average difference in attack performance between GraphPrivatizer and the baseline across all models and privacy budgets. We report p-values $P$ for a paired Wilcoxon signed rank test, testing respectively the null hypotheses $H_0$: $\text{Acc}_{\text{GP}} - \text{Acc}_{\text{RR}} = 0$ and $H_0$: $\text{AUC}_{\text{GP}} - \text{AUC}_{\text{RR}} = 0$. In sub-table (b) GAP denotes the utility gap, i.e., the accuracy loss with respect to the non-edge-private, $\epsilon = \infty$ setting corresponding to LPGNN (Sajadmanesh & Gatica-Perez, 2021), where we report aggregate results across models as (mean $\pm$ standard deviation), for $\epsilon = 0.1$.

(a) Accuracy and AUC results across all privacy budgets.

| dataset | $\text{Acc}_{\text{GP}} - \text{Acc}_{\text{RR}}$ | $\text{AUC}_{\text{GP}} - \text{AUC}_{\text{RR}}$ |
|---|---|---|
| Cora | 3.9 $(P<.001)$ | 0.2 $(P>.05)$ |
| LastFM | 10.1 $(P<.001)$ | 0.5 $(P>.05)$ |
| PubMed | 0.8 $(P<.001)$ | 2.7 $(P>.05)$ |
| Facebook | 5.0 $(P<.001)$ | 0.6 $(P>.05)$ |
| Amazon Photo | 5.5 $(P<.01)$ | 3.9 $(P>.05)$ |

(b) Utility gap (GAP) for $\epsilon = 0.1$.

| dataset | $\text{GAP}_{\text{GP}}$ | $\text{GAP}_{\text{RR}}$ |
|---|---|---|
| Cora | 6.3 $\pm$ 2.2 | 12.4 $\pm$ 6.3 |
| LastFM | 6.3 $\pm$ 3.2 | 21.7 $\pm$ 13.8 |
| PubMed | 1.9 $\pm$ 0.2 | 2.7 $\pm$ 1.7 |
| Facebook | 5.0 $\pm$ 0.6 | 13.2 $\pm$ 8.5 |
| Amazon Photo | 3.4 $\pm$ 2.7 | 8.0 $\pm$ 3.3 |

Across all models and datasets, GraphPrivatizer provides an average 6.6 AUC points improvement in privacy with respect to the non-edge-private LPGNN (Sajadmanesh & Gatica-Perez, 2021) setting while suffering a 4.9% decrease in accuracy, while the RR baseline provides a similar 6.4 AUC points improvement in privacy but with a much less desirable 12.5% decrease in accuracy. Overall, GraphPrivatizer achieves thus a better privacy-utility trade-off than RR. Additional results in Appendix C.

## 5.2 Results for GraphPrivatizer and discussion

While in Section 5.1 we show that GraphPrivatizer outperforms the RR baseline approach and improves upon the privacy-utility trade-off, here we investigate our approach in more detail and determine to which degree the use of thresholds and aggregations described in Section 4 is beneficial. We are interested in establishing if positive threshold and aggregation parameters (i.e., $\alpha, \delta > 0$) consistently provide a better privacy-utility trade-off than the reference case where no aggregation or threshold are considered (i.e., $\alpha = \delta = 0$).

Table 2: Aggregate results for GraphPrivatizer. We denote as $\text{Acc}_{\text{GP}}$ and $\text{AUC}_{\text{GP}}$ the average accuracy and AUC results for GraphPrivatizer with $\alpha, \delta > 0$, where the average is taken across all positive tested values of $\alpha$ and $\delta$. We denote with $\overline{\Delta_{\text{Acc}}}$ the average accuracy difference with the $\alpha = \delta = 0$ case across all tested values of $\alpha$ and $\delta$ larger than zero, where positive values of $\overline{\Delta_{\text{Acc}}}$ indicate that GraphPrivatizer performs better when some thresholds are applied and/or aggregation is performed during the neighbor perturbation. Analogously, we denote with $\overline{\Delta_{\text{AUC}}}$ the average $\text{AUC} \times 10^2$ difference, where values close to zero indicate that GraphPrivatizer offers the same protection against privacy attack when introducing a positive threshold and/or feature aggregation during the neighbor perturbation. We denote the datasets we use as Cora (Cr), LastFM (FM), PubMed (PM), Facebook (Fb), and Amazon Photo (Ph). Results reported as (average values $\pm$ standard deviation), for $\epsilon = 0.1$.

| | | GP-t | | | | | | GP-m | | | |
|---|---|---|---|---|---|---|---|---|---|---|---|
| Model | Dataset | $\text{Acc}_{\text{GP}}$ | $\text{AUC}_{\text{GP}}$ | $\overline{\Delta_{\text{Acc}}}$ | $\overline{\Delta_{\text{AUC}}}$ | Model | Dataset | $\text{Acc}_{\text{GP}}$ | $\text{AUC}_{\text{GP}}$ | $\overline{\Delta_{\text{Acc}}}$ | $\overline{\Delta_{\text{AUC}}}$ |
| GAT | Cr | $78.0 \pm 1.7$ | $92 \pm 1$ | $\mathbf{7 \pm 3}$ | $\mathbf{3 \pm 4}$ | GAT | Cr | $78.9 \pm 1.8$ | $73 \pm 6$ | $\mathbf{3 \pm 2}$ | $0 \pm 3$ |
| | FM | $81.1 \pm 3.9$ | $57 \pm 2$ | $\mathbf{6 \pm 3}$ | $3 \pm 2$ | | FM | $82.1 \pm 3.0$ | $57 \pm 1$ | $2 \pm 2$ | $1 \pm 1$ |
| | PM | $82.0 \pm 0.3$ | $70 \pm 5$ | $\mathbf{1.5 \pm 0.7}$ | $4 \pm 2$ | | PM | $82.1 \pm 0.3$ | $72 \pm 5$ | $\mathbf{1.5 \pm 0.7}$ | $0 \pm 1$ |
| | Fb | $89.5 \pm 0.7$ | $63 \pm 4$ | $\mathbf{4 \pm 2}$ | $\mathbf{2 \pm 2}$ | | Fb | $90.3 \pm 0.7$ | $64 \pm 5$ | $\mathbf{2 \pm 1}$ | $0 \pm 1$ |
| | Ph | $83.3 \pm 1.7$ | $79 \pm 4$ | $\mathbf{11 \pm 5}$ | $9 \pm 6$ | | Ph | $85.9 \pm 3.8$ | $81 \pm 5$ | $0.5 \pm 0.9$ | $0 \pm 1$ |
| GCN | Cr | $79.7 \pm 1.1$ | $72 \pm 5$ | $\mathbf{4 \pm 2}$ | $8 \pm 5$ | GCN | Cr | $80.4 \pm 1.1$ | $93 \pm 0.1$ | $\mathbf{4 \pm 2}$ | $5 \pm 3$ |
| | FM | $85.7 \pm 0.7$ | $90 \pm 2$ | $\mathbf{3 \pm 1}$ | $10 \pm 5$ | | FM | $85.7 \pm 0.4$ | $92 \pm 2$ | $\mathbf{3 \pm 1}$ | $5 \pm 3$ |
| | PM | $82.0 \pm 0.3$ | $96 \pm 1$ | $\mathbf{1.4 \pm 0.7}$ | $10 \pm 5$ | | PM | $82.0 \pm 0.3$ | $97 \pm 1$ | $\mathbf{1.5 \pm 0.7}$ | $2 \pm 2$ |
| | Fb | $90.4 \pm 0.2$ | $96 \pm 1$ | $\mathbf{4 \pm 2}$ | $6 \pm 3$ | | Fb | $91.3 \pm 0.2$ | $98 \pm 1$ | $\mathbf{1.5 \pm 0.7}$ | $2 \pm 1$ |
| | Ph | $81.4 \pm 1.9$ | $95 \pm 1$ | $\mathbf{13 \pm 7}$ | $3 \pm 3$ | | Ph | $83.1 \pm 3.8$ | $96 \pm 1$ | $\mathbf{1.2 \pm 1.0}$ | $2 \pm 4$ |
| SAGE | Cr | $79.7 \pm 1.0$ | $78 \pm 2$ | $\mathbf{4 \pm 2}$ | $5 \pm 3$ | SAGE | Cr | $80.4 \pm 1.0$ | $77 \pm 3$ | $\mathbf{4 \pm 2}$ | $3 \pm 2$ |
| | FM | $83.5 \pm 2.0$ | $85 \pm 3$ | $\mathbf{3 \pm 2}$ | $1 \pm 1$ | | FM | $84.2 \pm 1.3$ | $84 \pm 3$ | $\mathbf{3 \pm 1}$ | $1 \pm 1$ |
| | PM | $81.7 \pm 0.3$ | $56 \pm 3$ | $\mathbf{1.3 \pm 0.7}$ | $0 \pm 1$ | | PM | $81.6 \pm 0.3$ | $55 \pm 3$ | $\mathbf{1.5 \pm 0.7}$ | $0 \pm 1$ |
| | Fb | $90.4 \pm 0.2$ | $74 \pm 02$ | $\mathbf{4 \pm 2}$ | $4 \pm 2$ | | Fb | $90.9 \pm 0.2$ | $74 \pm 2$ | $\mathbf{2 \pm 1}$ | $1 \pm 2$ |
| | Ph | $83.4 \pm 0.6$ | $85 \pm 2$ | $\mathbf{13 \pm 5}$ | $17 \pm 9$ | | Ph | $86.1 \pm 3.5$ | $90 \pm 2$ | $\mathbf{0.8 \pm 0.6}$ | $8 \pm 5$ |

We present an aggregate overview of our results in Table 2. We focus here on the case $\epsilon = 0.1$ and on the GCN, GraphSAGE, and GAT models, with additional results available in Appendix C. GraphPrivatizer consistently performs better with $\alpha, \delta > 0$ in terms of accuracy, as it provides a positive accuracy improvement $\overline{\Delta_{\text{Acc}}}$ in almost the totality of cases. In more than half of the cases, it also provides performance in defending against privacy attacks which is equivalent to the $\alpha = \delta = 0$ setting, having $\overline{\Delta_{\text{AUC}}}$ which overlaps with zero. It should moreover be noted that the $\overline{\Delta_{\text{AUC}}}$ results reported in Table 2 are, as mentioned, multiplied by a factor $10^2$. For this reason, most of the cases where $\overline{\Delta_{\text{AUC}}} > 0$ correspond to only a small decrease in privacy, with the majority of cases reporting $\overline{\Delta_{\text{AUC}}} \leq 2$ which indicates an increase in the likelihood that the attacker is able to correctly identify edges of at most 2%. That is, it is on average preferable to introduce feature aggregation and/or to a threshold to select similar neighbors during the neighbor perturbation. If we compare the different models tested, GCN appears to be more prone to worse performance against privacy attacks. This behavior is consistent with the observations reported in Wu et al. (2022) and may be explained by highlighting how the influence computation that underlies the LinkTeller attack is better suited for graph convolution aggregations, and thus for GCNs (more details on Wu et al. (2022)). While information propagation between nodes can be exploited for other GNN architecture the introduction of, e.g., the attention mechanism in GAT/GATv2 can negatively impact the attack performance. Nevertheless, Table 1 and Table 2 show that GraphPrivatizer performs well for different GNN models and datasets.

While Table 2 shows that the setting $\alpha, \delta > 0$ is generally preferable when considering averaged results across all values of $\alpha$ and $\delta$, a more detailed analysis shows that, depending on the model and the dataset, specific combinations of $\alpha$ and $\delta$ provide the best performance. We report here more detailed results for GP-t with GAT, leaving additional results in Appendix C.

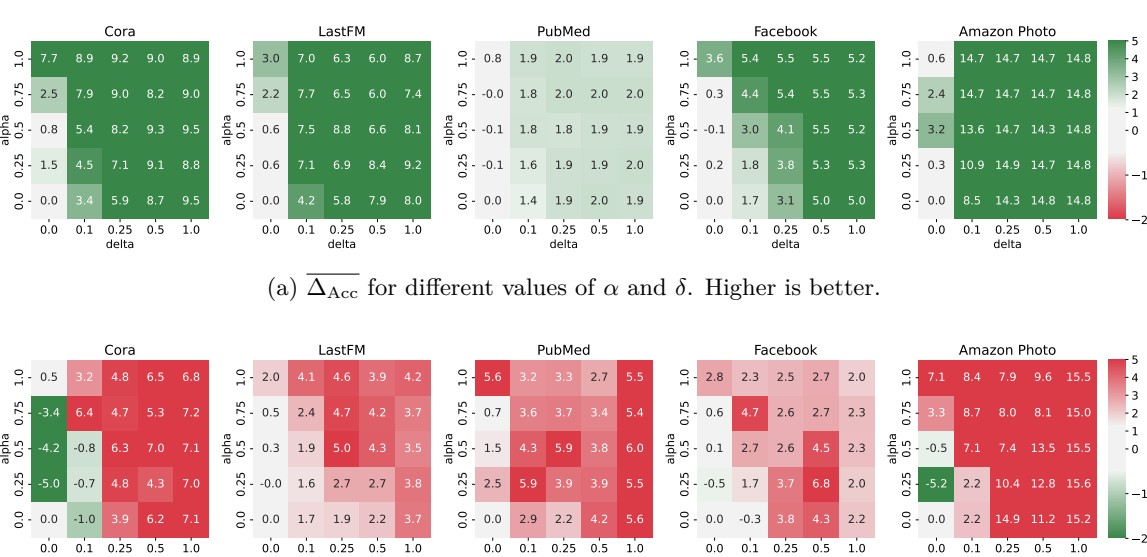

(a) $\overline{\Delta_{\text{Acc}}}$ for different values of $\alpha$ and $\delta$. Higher is better.

(b) $\overline{\Delta_{\text{AUC}}}$ for different values of $\alpha$ and $\delta$. Smaller is better.

Figure 3: Results for GP-t with GAT, for $\epsilon = 0.1$. Average values across 10 runs. The colormap is normalized to interpret all $\overline{\Delta_{\text{Acc}}} > 0$ and $\overline{\Delta_{\text{AUC}}} \leq 2$ as desirable.

Figure 3 exemplifies the trade-off between improvements in model accuracy, and a decrease in edge privacy for all combinations of $\alpha$ and $\delta$, with the combination $\alpha = \delta = 0$ representing the reference base case. Model accuracy tends to increase with larger values of $\alpha$ and $\delta$, while privacy tends to decrease. In fact, a higher threshold $\delta$ implies that fewer nodes may be selected as candidate replacements during perturbation, with $\delta = 1$ requiring that only nodes with similarity of 1 may be considered. With respect to the aggregation parameter $\alpha$, values larger than zero tend to provide increased accuracy for small $\delta$. This suggests that, when computing similarities, it is beneficial to do so on aggregate features: in this way, nodes which behave similarly with respect to the output of the AGGREGATE function (Section 3.1), which is then used to train the GNN, will be favoured as candidates for replacement.

Small values of $\alpha$ and $\delta$ provide therefore a good trade-off between model accuracy and privacy, with some combinations remarkably offering both improved accuracy and better privacy over the $\alpha = \delta = 0$ case. For instance, the combination $\alpha = 0.5$, $\delta = 0.1$ for Cora offers a 5.4% improvement in accuracy *and* a 0.8% improvement in edge privacy. Considering the additional visualizations provided on Appendix C, this behavior generally holds across datasets, for both GP-t and GP-m. Even for GCN (see for instance Table 3b), where average results show an unfavorable $\overline{\Delta_{\text{AUC}}} > 0$, specific combinations of $\alpha$ and $\delta$ can provide accuracy improvements over the $\alpha = \delta = 0$ case at a small or null privacy cost.

Finally, we consider what the benefits of GraphPrivatizer are across the range of privacy budgets we tested. We focus here on GP-t on LastFM for GCN: Table 2 shows for this case a decrease in protection against privacy attacks corresponding to a relatively large $\overline{\Delta_{\text{AUC}}} = 10 \pm 5$ when introducing a positive $\alpha$ or $\delta$.

Analyzing this case more in detail, Table 3a shows that, as previously observed, on average GraphPrivatizer entails a 10% increase in the attack performance for this specific experiment for positive thresholds or aggregation coefficients. Nevertheless, $\text{Acc}_{\text{GP}}$ for $\epsilon = 0.1$ is close to that of the non private model for $\epsilon = \infty$, despite having a smaller AUC. More in detail, when compared to GraphPrivatizer for $\epsilon = 0.1$, the no aggregations and no thresholds achieve a better accuracy only for $\epsilon = 8$, but with a worse AUC. That is, on average, GraphPrivatizer with threshold and feature aggregation provides a better trade-off between

Table 3: Accuracy and AUC across different values of $\epsilon$. We denote as with the subscript GP the average results for GraphPrivatizer across $\alpha, \delta > 0$, while the subscript b denotes the base case $\alpha = \delta = 0$.

(a) GCN on LastFM with GP-t. Average across $\alpha, \delta$.

| $\epsilon$ | $\overline{\Delta_{\text{Acc}}}$ | $\text{Acc}_{\text{GP}}$ | $\text{Acc}_{\text{b}}$ | $\text{AUC}_{\text{GP}}$ | $\text{AUC}_{\text{b}}$ |
|---|---|---|---|---|---|
| 0.1 | $3.2 \pm 1.0$ | $85.7 \pm 1.5$ | 82.6 | $90 \pm 5$ | 80 |
| 1 | $1.7 \pm 1.0$ | $85.9 \pm 1.1$ | 84.2 | $92 \pm 3$ | 84 |
| 2 | $1.7 \pm 1.0$ | $86.2 \pm 0.7$ | 84.5 | $93 \pm 2$ | 89 |
| 8 | $0.1 \pm 0.0$ | $86.6 \pm 0.1$ | 86.4 | $94 \pm 0$ | 94 |
| $\infty$ | $0.0 \pm 0.0$ | $86.5 \pm 0.0$ | 86.5 | $94 \pm 0$ | 94 |

(b) GCN on LastFM with GP-t. $\alpha = 0.75$, $\delta = 0$.

| $\epsilon$ | $\overline{\Delta_{\text{Acc}}}$ | $\text{Acc}_{\text{GP}}$ | $\text{Acc}_{\text{b}}$ | $\text{AUC}_{\text{GP}}$ | $\text{AUC}_{\text{b}}$ |
|---|---|---|---|---|---|
| 0.1 | 0.4 | 83.0 | 82.6 | 80 | 81 |
| 1 | 1.1 | 84.1 | 83.0 | 85 | 85 |
| 2 | 0.6 | 85.1 | 84.5 | 89 | 89 |
| 8 | 0.2 | 86.6 | 86.4 | 94 | 95 |
| $\infty$ | 0 | 86.6 | 86.6 | 95 | 95 |

accuracy and privacy. Furthermore, the improvement is more evident for specific values of $\alpha$ and $\delta$: for instance, Table 3b shows that for $\epsilon = 0.1$, $\alpha = 0.75$ and $\delta = 0$, provide both a 0.4% improvement in accuracy *and* a 1% improvement in AUC. Similarly, higher privacy budgets are still favorable to GraphPrivatizer with $\alpha, \delta > 0$ which always outperforms the $\alpha = \delta = 0$ case at a smaller or equal AUC.

Considering existing results available in literature, Sajadmanesh et al. (2023) perform experiments on the Facebook dataset and obtain an accuracy of $76.3 \pm 0.21$ for a total privacy budget of $\epsilon = 4$. This result is, however, not directly comparable to ours as it considers a different privacy setting (global vs local privacy) and assumes access to the adjacency matrix. Closer to our approach is that of Joshi & Mishra (2023) who train an edge-LDP GNN where nodes have noiseless access to part of the adjacency matrix: despite the less strict privacy setting, they obtain worse accuracy and report a best accuracy of $\approx 50$ on Cora, $\approx 70$ on LastFM, and $\approx 70$ on PubMed, while not testing the effectiveness of their method against privacy attacks that try to recover the edges. Their approach is, indeed, very similar to the one we adopt in Section 5.1 as a baseline which we have shown is consistently outperformed by our method and provides generally worse privacy-utility trade-off across all datasets.

## 6 Conclusions

Motivated by real-world applications, we investigated LDP GNNs. Considering a local privacy setting where the individual nodes of the graph can only have noise-free access to their own features, labels, and edges, we (i) introduced a new definition of LDP for our local privacy setting, (ii) proposed our private algorithm GraphPrivatizer, and (iii) empirically validated its performance on real-world datasets and against privacy attacks. GraphPrivatizer is a fully private algorithm that protects edges, features, and labels. We introduced a new methodology to protect edges by means of controlled perturbations which replace the neighbors of a node with other similar nodes according to some similarity measure. We furthermore evaluated the impact of thresholds on the similarity between the original and the perturbed neighbor nodes and of feature aggregation in computing the similarity scores, finding that a positive threshold and aggregation coefficient provide the best privacy-utility trade-off. Compared to existing approaches which do not provide edge-privacy (Sajadmanesh & Gatica-Perez, 2021) or do so while requiring a central entity to have complete information about the edges of the graph (Sajadmanesh et al., 2023; Joshi & Mishra, 2022), GraphPrivatizer provides LDP without requiring a trusted entity to have access to the adjacency matrix of the graph and is applicable to a variety of GNN models.

Future work could focus on addressing some of the limitations of GraphPrivatizer. First, GraphPrivatizer does not consider datasets containing edge features. While it could be possible to, e.g., privatize categorical edge features using randomized response, our local privacy setting would need to be adapted to determine which entity can have noise-free access to the edge features. Additionally, better personalized user privacy requirements could be considered. In particular, the introduction of a notion of *trust* between nodes could allow two neighboring nodes that trust each other to exchange less noisy information, thus improving the GNN performance. Finally, while GraphPrivatizer is generally applicable to various types of GNN models, edge perturbations which are tailored to a specific GNN architecture can possibly provide a better privacy-utility trade-off and could therefore be explored in future work.

**Acknowledgments**

We would like to thank Prof. Thomas Gärtner (TU Wien) for his helpful comments on the manuscript. This work was funded in part by the TU Wien DK SecInt and by the Austrian Science Fund (FWF), project NanOX-ML (6728). The computational results presented have been achieved in part using the Vienna Scientific Cluster (VSC) and NISER:RIN4001.

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

## A   Experimental details

For all experiments, we divide our dataset with 50:25:25 train:validation:test set ratios, similarly to Sajad-manesh & Gatica-Perez (2021). We train all the models for 100 epochs, with learning rate $10^{-2}$, weight decay $10^{-3}$, and dropout rate 0.5. We run experiments with edge privacy budget $\epsilon = \{0.1, 1, 2, 8, \infty\}$, $\alpha = \{0, 0.25, 0.5, 0.75, 1.0\}$, and $\delta = \{0, 0.1, 0.25, 0.5, 1\}$. We perform 10 runs for each value of $\epsilon$, $\alpha$ and $\delta$ with different seeds and consider average results. We perform label and feature perturbations as described in Section 4.3, with privacy budgets fixed at $\epsilon_x = 3$ and $\epsilon_y = 3$. Additionally, we set the KProp hyper-parameters in the Drop algorithm to the best values described in Sajadmanesh & Gatica-Perez (2021): we use $K_x = 16$, $K_y = 2$ for Cora, $K_x = 4$, $K_y = 2$ for Facebook, and $K_x = 16$, $K_y = 0$ for LastFM and PubMed. After a grid-search tuning, we use $K_x = 4$, $K_y = 2$ for Amazon Photo.

Table 4: Statistics of the datasets. Cora and PubMed Yang et al. (2016) are citation networks where an edge $(i, j)$ exists between two documents if document $i$ cites document $j$, and features consist of bag-of-words representations of the documents. Classes consist of document categories. Facebook Rozemberczki et al. (2021) is a page-page graph of verified Facebook pages, where nodes correspond to official Facebook pages and edges correspond to mutual likes between pages. Node features are extracted from the site descriptions and the classes correspond to various page categories. LastFM Rozemberczki & Sarkar (2020) is a friendship graph of LastFM users where nodes represent users and edges friendships between users. The classes correspond to the home countries of the users. Amazon Photo (Shchur et al., 2018) is a co-purchase network where nodes represent goods and edges correspond to two goods which have been frequently bought together on Amazon. The features consist of bag-of-words representations of the goods and the classes correspond to product categories.

| Dataset | Classes | Nodes | Edges | Features | Avg. Degree |
|---|---|---|---|---|---|
| Cora (Cr) | 7 | 2708 | 5278 | 1433 | 3.90 |
| LastFM (FM) | 10 | 7083 | 25814 | 7842 | 7.29 |
| PubMed (PM) | 3 | 19717 | 44324 | 500 | 4.50 |
| Facebook (Fb) | 4 | 22470 | 170912 | 4714 | 15.21 |
| Amazon Photo (Ph) | 8 | 7650 | 238162 | 745 | 31.13 |

## B   Similarity metric

The performance of GraphPrivatizer is affected by the choice of similarity measure in Algorithm 2. Non-parametric and non-learnable similarity measures are desirable in our local privacy setting, as they do not require access to data. We compare the cosine similarity with the Euclidean distance, and find that the cosine similarity is preferable. Refer to Section 5 for details on how privacy attacks are performed.

Table 5: We denote with $\overline{\Delta_{\text{Acc}}}$ the average accuracy difference between results obtained with the cosine similarity and with the Euclidean distance. Analogously, we denote with $\overline{\Delta_{\text{AUC}}}$ the average AUC $\times 10^2$ difference. Results reported as (average $\pm$ standard deviation), for $\epsilon = 0.1$ across 5 runs. The cosine similarity is preferable as it provides better accuracy ($\overline{\Delta_{\text{Acc}}} > 0$) with comparable privacy ($\overline{\Delta_{\text{AUC}}} \approx 0$).

| Dataset | $\overline{\Delta_{\text{Acc}}}$ | $\overline{\Delta_{\text{AUC}}}$ |
|---|---|---|
| Cora | $4.8 \pm 1.2$ | $2 \pm 3$ |
| Facebook | $1.6 \pm 0.5$ | $1 \pm 1$ |
| LastFM | $2.4 \pm 1.4$ | $1 \pm 1$ |
| Amazon Photo | $-0.9 \pm 5$ | $-1 \pm 10$ |
| Pubmed | $2.8 \pm 0.3$ | $3 \pm 7$ |

# C   Additional results

## C.1   Comparison against the RR baseline

Table 6: Results on all datasets for GraphPrivatizer and the RR baseline discussed in Section 5.1. The subscript $\infty$ denotes the non-edge-private results.

| Dataset | Model | $\Delta_{\text{Acc}}$ | $\Delta_{\text{AUC}}$ | $\text{Acc}_{\text{GP}}$ | $\text{Acc}_{\text{RR}}$ | $\text{AUC}_{\text{GP}}$ | $\text{AUC}_{\text{RR}}$ | $\text{Acc}_\infty$ | $\text{AUC}_\infty$ |
|---|---|---|---|---|---|---|---|---|---|
| Cora | GAT | 3.2 | −1.1 | $71.9 \pm 2.6$ | $68.8 \pm 5.4$ | $65.0 \pm 4.8$ | $66.0 \pm 7.9$ | $80.0 \pm 1.7$ | $78.5 \pm 6.9$ |
| | GATv2 | 1.2 | 1.2 | $71.0 \pm 2.3$ | $69.8 \pm 2.3$ | $57.0 \pm 2.8$ | $55.8 \pm 3.1$ | $79.7 \pm 2.0$ | $61.6 \pm 7.4$ |
| | GCN | 3.1 | −8.8 | $75.7 \pm 1.0$ | $72.6 \pm 2.4$ | $83.4 \pm 0.9$ | $92.2 \pm 2.0$ | $81.7 \pm 1.0$ | $95.7 \pm 0.9$ |
| | GConv | 22.2 | 5.8 | $71.1 \pm 2.9$ | $48.9 \pm 6.4$ | $56.6 \pm 1.3$ | $50.8 \pm 0.5$ | $74.0 \pm 4.9$ | $53.9 \pm 1.4$ |
| | GT | 3.4 | 4.1 | $71.6 \pm 2.9$ | $68.3 \pm 2.6$ | $58.0 \pm 0.9$ | $53.8 \pm 0.8$ | $79.2 \pm 2.8$ | $62.3 \pm 1.1$ |
| | SAGE | 3.4 | −5.1 | $77.0 \pm 1.0$ | $73.6 \pm 2.1$ | $72.0 \pm 3.3$ | $77.1 \pm 2.8$ | $81.8 \pm 0.7$ | $79.3 \pm 3.1$ |
| Facebook | GAT | 4.2 | 5.1 | $85.7 \pm 0.9$ | $81.5 \pm 0.5$ | $60.8 \pm 3.7$ | $55.7 \pm 3.3$ | $90.9 \pm 0.6$ | $64.6 \pm 3.7$ |
| | GATv2 | 3.4 | −0.7 | $83.0 \pm 1.0$ | $79.5 \pm 1.1$ | $53.7 \pm 1.7$ | $54.4 \pm 3.5$ | $88.6 \pm 2.1$ | $61.3 \pm 5.9$ |
| | GCN | 5.9 | −1.6 | $86.5 \pm 0.3$ | $80.6 \pm 0.4$ | $91.3 \pm 0.9$ | $92.9 \pm 1.1$ | $91.9 \pm 0.2$ | $99.1 \pm 0.4$ |
| | GConv | 26.7 | 2.3 | $84.6 \pm 0.8$ | $57.9 \pm 8.8$ | $52.4 \pm 0.9$ | $50.1 \pm 0.1$ | $88.5 \pm 0.9$ | $51.1 \pm 0.5$ |
| | GT | 3.9 | 1.5 | $86.7 \pm 0.5$ | $82.8 \pm 0.4$ | $53.8 \pm 0.7$ | $52.2 \pm 0.7$ | $91.7 \pm 0.1$ | $60.3 \pm 1.7$ |
| | SAGE | 5.3 | −0.8 | $87.0 \pm 0.2$ | $81.7 \pm 0.4$ | $69.7 \pm 2.0$ | $70.5 \pm 3.5$ | $91.9 \pm 0.2$ | $74.7 \pm 2.3$ |
| LastFM | GAT | 24.2 | 2.9 | $75.7 \pm 3.2$ | $51.5 \pm 14.8$ | $54.7 \pm 1.2$ | $51.8 \pm 1.2$ | $83.6 \pm 3.3$ | $57.0 \pm 2.2$ |
| | GATv2 | 9.8 | 1.5 | $68.9 \pm 8.9$ | $59.1 \pm 18.6$ | $52.5 \pm 0.7$ | $50.9 \pm 0.7$ | $77.3 \pm 8.3$ | $55.4 \pm 1.2$ |
| | GCN | 3.3 | −5.9 | $82.2 \pm 1.9$ | $78.9 \pm 2.3$ | $81.7 \pm 2.0$ | $87.6 \pm 3.1$ | $86.5 \pm 0.3$ | $94.8 \pm 1.9$ |
| | GConv | 42.3 | 2.8 | $65.2 \pm 6.9$ | $22.9 \pm 12.2$ | $52.9 \pm 1.7$ | $50.1 \pm 0.2$ | $66.6 \pm 10.8$ | $51.8 \pm 1.6$ |
| | GT | 8.1 | 3.4 | $69.0 \pm 4.4$ | $60.9 \pm 9.1$ | $54.8 \pm 0.7$ | $51.5 \pm 0.3$ | $79.1 \pm 5.1$ | $59.3 \pm 1.6$ |
| | SAGE | 4.8 | −1.2 | $80.5 \pm 1.7$ | $75.7 \pm 2.4$ | $82.8 \pm 3.5$ | $84.0 \pm 5.9$ | $86.0 \pm 0.6$ | $86.1 \pm 2.4$ |
| PubMed | GAT | 0.5 | 4.3 | $80.4 \pm 0.4$ | $79.9 \pm 0.4$ | $68.7 \pm 3.8$ | $64.5 \pm 5.8$ | $82.5 \pm 0.2$ | $72.8 \pm 5.6$ |
| | GATv2 | 0.2 | 2.8 | $80.7 \pm 0.4$ | $80.5 \pm 0.3$ | $59.1 \pm 2.5$ | $56.3 \pm 3.2$ | $82.6 \pm 0.2$ | $66.0 \pm 4.7$ |
| | GCN | 0.2 | −12.3 | $80.6 \pm 0.5$ | $80.3 \pm 0.4$ | $85.7 \pm 1.0$ | $98.1 \pm 0.5$ | $82.4 \pm 0.3$ | $99.2 \pm 0.7$ |
| | GConv | 3.9 | 20.5 | $79.8 \pm 0.6$ | $75.9 \pm 1.3$ | $71.3 \pm 3.9$ | $50.7 \pm 0.4$ | $82.0 \pm 0.5$ | $75.0 \pm 2.2$ |
| | GT | −0.7 | 7.7 | $80.0 \pm 0.6$ | $80.7 \pm 0.4$ | $60.5 \pm 0.5$ | $52.8 \pm 0.6$ | $82.2 \pm 0.3$ | $64.3 \pm 1.2$ |
| | SAGE | 0.1 | −1.5 | $80.3 \pm 0.3$ | $80.3 \pm 0.5$ | $56.9 \pm 3.1$ | $58.4 \pm 3.8$ | $82.0 \pm 0.2$ | $56.6 \pm 1.7$ |
| Amazon Photo | GAT | 2.5 | 11.6 | $82.7 \pm 3.5$ | $80.2 \pm 1.1$ | $72.9 \pm 3.4$ | $61.3 \pm 4.0$ | $86.8 \pm 0.6$ | $84.6 \pm 3.1$ |
| | GATv2 | 9.0 | 8.8 | $83.5 \pm 0.5$ | $74.4 \pm 4.2$ | $63.5 \pm 1.7$ | $54.7 \pm 1.6$ | $85.0 \pm 0.4$ | $76.8 \pm 5.1$ |
| | GCN | 8.1 | 3.6 | $80.4 \pm 6.5$ | $72.3 \pm 1.7$ | $95.1 \pm 0.7$ | $91.4 \pm 0.2$ | $80.8 \pm 9.1$ | $96.3 \pm 1.7$ |
| | GConv | −3.6 | −0.7 | $55.7 \pm 10.1$ | $59.3 \pm 1.3$ | $52.7 \pm 1.6$ | $53.4 \pm 2.1$ | $61.3 \pm 18.9$ | $52.2 \pm 1.3$ |
| | GT | 7.3 | 12.1 | $84.4 \pm 0.6$ | $77.1 \pm 2.4$ | $68.5 \pm 1.0$ | $56.4 \pm 3.7$ | $86.7 \pm 0.3$ | $87.4 \pm 1.8$ |
| | SAGE | 9.5 | −11.8 | $85.0 \pm 0.4$ | $75.5 \pm 1.8$ | $78.9 \pm 5.1$ | $90.6 \pm 0.7$ | $86.4 \pm 0.2$ | $93.8 \pm 1.0$ |

## C.2 Additional results for the GT, GATv2, and GConv models

Table 7: Aggregate results for GraphPrivatizer for all models. We denote as $\text{Acc}_{\text{GP}}$ and $\text{AUC}_{\text{GP}}$ the average accuracy and AUC results for GraphPrivatizer with $\alpha, \delta > 0$, where the average is taken across all positive tested values of $\alpha$ and $\delta$. We denote with $\overline{\Delta_{\text{Acc}}}$ the average accuracy difference with the $\alpha = \delta = 0$ case across all tested values of $\alpha$ and $\delta$ larger than zero, where positive values of $\overline{\Delta_{\text{Acc}}}$ indicate that GraphPrivatizer performs better when some thresholds are applied and/or aggregation is performed during the neighbor perturbation. Analogously, we denote with $\overline{\Delta_{\text{AUC}}}$ the average AUC$\times 10^2$ difference, where values close to zero indicate that GraphPrivatizer offers the same protection against privacy attack when introducing a positive threshold and/or feature aggregation during the neighbor perturbation. We denote the datasets we use as Cora (Cr), LastFM (FM), PubMed (PM), Facebook (Fb), and Amazon Photo (Ph). Results reported as (average values $\pm$ standard deviation), for $\epsilon = 0.1$.

| GP-t | | | | | | GP-m | | | | | |
|---|---|---|---|---|---|---|---|---|---|---|---|
| Model | Dataset | $\text{Acc}_{\text{GP}}$ | $\text{AUC}_{\text{GP}}$ | $\overline{\Delta_{\text{Acc}}}$ | $\overline{\Delta_{\text{AUC}}}$ | Model | Dataset | $\text{Acc}_{\text{GP}}$ | $\text{AUC}_{\text{GP}}$ | $\overline{\Delta_{\text{Acc}}}$ | $\overline{\Delta_{\text{AUC}}}$ |
| GAT | Cr | $78.0 \pm 1.7$ | $92 \pm 1$ | $\mathbf{7 \pm 3}$ | $\mathbf{3 \pm 4}$ | GAT | Cr | $78.9 \pm 1.8$ | $73 \pm 6$ | $\mathbf{3 \pm 2}$ | $\mathbf{0 \pm 3}$ |
| | FM | $81.1 \pm 3.9$ | $57 \pm 2$ | $\mathbf{6 \pm 3}$ | $3 \pm 2$ | | FM | $82.1 \pm 3.0$ | $57 \pm 1$ | $2 \pm 2$ | $1 \pm 1$ |
| | PM | $82.0 \pm 0.3$ | $70 \pm 5$ | $\mathbf{1.5 \pm 0.7}$ | $4 \pm 2$ | | PM | $82.1 \pm 0.3$ | $72 \pm 5$ | $\mathbf{1.5 \pm 0.7}$ | $0 \pm 1$ |
| | Fb | $89.5 \pm 0.7$ | $63 \pm 4$ | $\mathbf{4 \pm 2}$ | $\mathbf{2 \pm 2}$ | | Fb | $90.3 \pm 0.7$ | $64 \pm 5$ | $\mathbf{2 \pm 1}$ | $0 \pm 1$ |
| | Ph | $83.3 \pm 1.7$ | $79 \pm 4$ | $\mathbf{11 \pm 5}$ | $9 \pm 6$ | | Ph | $85.9 \pm 3.8$ | $81 \pm 5$ | $0.5 \pm 0.9$ | $0 \pm 1$ |
| GCN | Cr | $79.7 \pm 1.1$ | $72 \pm 5$ | $\mathbf{4 \pm 2}$ | $8 \pm 5$ | GCN | Cr | $80.4 \pm 1.1$ | $93 \pm 0.1$ | $\mathbf{4 \pm 2}$ | $5 \pm 3$ |
| | FM | $85.7 \pm 0.7$ | $90 \pm 2$ | $\mathbf{3 \pm 1}$ | $10 \pm 5$ | | FM | $85.7 \pm 0.4$ | $92 \pm 2$ | $\mathbf{3 \pm 1}$ | $5 \pm 3$ |
| | PM | $82.0 \pm 0.3$ | $96 \pm 1$ | $\mathbf{1.4 \pm 0.7}$ | $10 \pm 5$ | | PM | $82.0 \pm 0.3$ | $97 \pm 1$ | $\mathbf{1.5 \pm 0.7}$ | $\mathbf{2 \pm 2}$ |
| | Fb | $90.4 \pm 0.2$ | $96 \pm 1$ | $\mathbf{4 \pm 2}$ | $6 \pm 3$ | | Fb | $91.3 \pm 0.2$ | $98 \pm 1$ | $\mathbf{1.5 \pm 0.7}$ | $2 \pm 1$ |
| | Ph | $81.4 \pm 1.9$ | $95 \pm 1$ | $\mathbf{13 \pm 7}$ | $\mathbf{3 \pm 3}$ | | Ph | $83.1 \pm 3.8$ | $96 \pm 1$ | $\mathbf{1.2 \pm 1.0}$ | $\mathbf{2 \pm 4}$ |
| SAGE | Cr | $79.7 \pm 1.0$ | $78 \pm 2$ | $\mathbf{4 \pm 2}$ | $5 \pm 3$ | SAGE | Cr | $80.4 \pm 1.0$ | $77 \pm 3$ | $\mathbf{4 \pm 2}$ | $3 \pm 2$ |
| | FM | $83.5 \pm 2.0$ | $85 \pm 3$ | $\mathbf{3 \pm 2}$ | $\mathbf{1 \pm 1}$ | | FM | $84.2 \pm 1.3$ | $84 \pm 3$ | $\mathbf{3 \pm 1}$ | $1 \pm 1$ |
| | PM | $81.7 \pm 0.3$ | $56 \pm 3$ | $\mathbf{1.3 \pm 0.7}$ | $\mathbf{0 \pm 1}$ | | PM | $81.6 \pm 0.3$ | $55 \pm 3$ | $\mathbf{1.5 \pm 0.7}$ | $0 \pm 1$ |
| | Fb | $90.4 \pm 0.2$ | $74 \pm 02$ | $\mathbf{4 \pm 2}$ | $4 \pm 2$ | | Fb | $90.9 \pm 0.2$ | $74 \pm 2$ | $\mathbf{2 \pm 1}$ | $1 \pm 2$ |
| | Ph | $83.4 \pm 0.6$ | $85 \pm 2$ | $\mathbf{13 \pm 5}$ | $17 \pm 9$ | | Ph | $86.1 \pm 3.5$ | $90 \pm 2$ | $\mathbf{0.8 \pm 0.6}$ | $8 \pm 5$ |
| GT | Cr | $78.1 \pm 1.7$ | $61 \pm 2$ | $\mathbf{6 \pm 3}$ | $3 \pm 2$ | GT | Cr | $78.4 \pm 1.4$ | $62 \pm 1$ | $\mathbf{4 \pm 2}$ | $1 \pm 1$ |
| | FM | $78.6 \pm 1.1$ | $54 \pm 1$ | $\mathbf{7 \pm 4}$ | $3 \pm 2$ | | FM | $79.6 \pm 2.9$ | $59 \pm 1$ | $\mathbf{3 \pm 1}$ | $1 \pm 1$ |
| | PM | $81.7 \pm 0.3$ | $64 \pm 1$ | $\mathbf{2 \pm 1}$ | $3 \pm 1$ | | PM | $81.7 \pm 0.3$ | $64 \pm 1$ | $\mathbf{2 \pm 1}$ | $1 \pm 1$ |
| | Fb | $90.5 \pm 0.3$ | $59 \pm 1$ | $\mathbf{4 \pm 2}$ | $5 \pm 2$ | | Fb | $90.1 \pm 0.3$ | $59 \pm 1$ | $\mathbf{3 \pm 1}$ | $1 \pm 1$ |
| | Ph | $84.3 \pm 1.3$ | $78 \pm 4$ | $\mathbf{7 \pm 4}$ | $13 \pm 8$ | | Ph | $86.2 \pm 3$ | $81 \pm 4$ | $\mathbf{1.0 \pm 0.7}$ | $5 \pm 7$ |
| GATv2 | Cr | $77.3 \pm 2.2$ | $60 \pm 3$ | $\mathbf{7 \pm 3}$ | $4 \pm 2$ | GATv2 | Cr | $77.8 \pm 1.9$ | $60 \pm 5$ | $\mathbf{5 \pm 2}$ | $2 \pm 2$ |
| | FM | $73.8 \pm 1.1$ | $54 \pm 1$ | $\mathbf{4 \pm 3}$ | $2 \pm 1$ | | FM | $76.2 \pm 8.2$ | $55 \pm 1$ | $1 \pm 2$ | $0 \pm 1$ |
| | PM | $82.2 \pm 0.4$ | $64 \pm 4$ | $\mathbf{1.2 \pm 0.7}$ | $3 \pm 2$ | | PM | $82.1 \pm 0.3$ | $65 \pm 4$ | $\mathbf{1.4 \pm 0.7}$ | $4 \pm 1$ |
| | Fb | $87.3 \pm 1.5$ | $58 \pm 4$ | $\mathbf{4 \pm 1}$ | $3 \pm 2$ | | Fb | $87.8 \pm 1.7$ | $59 \pm 4$ | $\mathbf{3 \pm 1}$ | $2 \pm 2$ |
| | Ph | $81.1 \pm 1.6$ | $68 \pm 5$ | $\mathbf{11 \pm 6}$ | $\mathbf{7 \pm 7}$ | | Ph | $82.3 \pm 2.6$ | $70 \pm 5$ | $2 \pm 2$ | $\mathbf{2 \pm 5}$ |
| GConv | Cr | $75.4 \pm 3.6$ | $55 \pm 2$ | $\mathbf{4 \pm 2}$ | $\mathbf{-2 \pm 1}$ | GConv | Cr | $75.5 \pm 3.1$ | $54 \pm 1$ | $1 \pm 1$ | $\mathbf{-2 \pm 1}$ |
| | FM | $69.9 \pm 5.7$ | $52 \pm 1$ | $\mathbf{3 \pm 2}$ | $\mathbf{-1 \pm 1}$ | | FM | $70.0 \pm 6.2$ | $53 \pm 2$ | $1 \pm 2$ | $\mathbf{-1 \pm 1}$ |
| | PM | $81.4 \pm 0.4$ | $74 \pm 3$ | $\mathbf{1.7 \pm 0.7}$ | $3 \pm 2$ | | PM | $81.5 \pm 0.4$ | $75 \pm 3$ | $\mathbf{1.2 \pm 0.7}$ | $\mathbf{-1 \pm 1}$ |
| | Fb | $87.7 \pm 0.8$ | $51 \pm 1$ | $\mathbf{4 \pm 2}$ | $\mathbf{-1 \pm 1}$ | | Fb | $88.1 \pm 0.8$ | $51 \pm 1$ | $\mathbf{3 \pm 1}$ | $\mathbf{0 \pm 0}$ |
| | Ph | $64.3 \pm 7.9$ | $51 \pm 1$ | $3 \pm 6$ | $\mathbf{0 \pm 1}$ | | Ph | $71.1 \pm 9.9$ | $51 \pm 2$ | $5 \pm 7$ | $\mathbf{0 \pm 1}$ |

## C.3 Results for different privacy budgets

We denote as with the subscript GP the average results for GraphPrivatizer across $\alpha, \delta > 0$, while the subscript b denotes the baseline $\alpha = \delta = 0$.

Table 8: Results on Cora.

| Model | $\epsilon$ | GP-t $\overline{\Delta_{Acc}}$ | $Acc_{GP}$ | $Acc_b$ | $AUC_{GP}$ | $AUC_b$ | GP-m $\overline{\Delta_{Acc}}$ | $Acc_{GP}$ | $Acc_b$ | $AUC_{GP}$ | $AUC_b$ |
|---|---|---|---|---|---|---|---|---|---|---|---|
| GAT | 0.1 | $8.2 \pm 1.0$ | $79.3 \pm 1.4$ | 71.1 | $74.2 \pm 2.5$ | 69.2 | $3.4 \pm 2.0$ | $79.8 \pm 1.6$ | 76.4 | $74.8 \pm 2.7$ | 73.2 |
|  | 1.0 | $4.2 \pm 1.0$ | $79.7 \pm 1.0$ | 75.6 | $73.6 \pm 3.1$ | 69.1 | $2.0 \pm 1.0$ | $80.3 \pm 0.7$ | 78.3 | $73.8 \pm 2.6$ | 70.6 |
|  | 2.0 | $1.9 \pm 1.0$ | $79.8 \pm 0.6$ | 77.8 | $74.2 \pm 2.2$ | 72.5 | $0.6 \pm 0.0$ | $80.7 \pm 0.4$ | 80.1 | $74.9 \pm 2.1$ | 74.9 |
|  | 8.0 | $-0.6 \pm 0.0$ | $80.0 \pm 0.4$ | 80.6 | $73.7 \pm 2.8$ | 78.7 | $-0.4 \pm 0.0$ | $80.9 \pm 0.4$ | 81.3 | $75.5 \pm 1.8$ | 73.6 |
|  | $\infty$ | $-0.1 \pm 0.0$ | $80.0 \pm 0.3$ | 80.2 | $77.3 \pm 1.4$ | 77.5 | $0.0 \pm 0.0$ | $81.0 \pm 0.2$ | 81.0 | $76.4 \pm 1.3$ | 73.3 |
| GATv2 | 0.1 | $8.2 \pm 1.0$ | $78.8 \pm 0.8$ | 70.6 | $61.1 \pm 1.1$ | 56.6 | $5.8 \pm 1.0$ | $79.0 \pm 1.0$ | 73.2 | $61.3 \pm 0.7$ | 58.6 |
|  | 1.0 | $4.6 \pm 0.0$ | $79.2 \pm 0.3$ | 74.5 | $61.6 \pm 0.8$ | 60.3 | $2.0 \pm 1.0$ | $79.1 \pm 0.5$ | 77.1 | $61.5 \pm 0.8$ | 60.1 |
|  | 2.0 | $2.3 \pm 1.0$ | $79.2 \pm 0.5$ | 77.0 | $62.3 \pm 1.3$ | 59.2 | $2.0 \pm 1.0$ | $79.3 \pm 0.7$ | 77.4 | $62.2 \pm 1.0$ | 61.1 |
|  | 8.0 | $0.5 \pm 0.0$ | $79.1 \pm 0.5$ | 78.6 | $62.0 \pm 0.9$ | 61.3 | $0.1 \pm 1.0$ | $79.2 \pm 0.6$ | 79.1 | $60.9 \pm 1.2$ | 62.2 |
|  | $\infty$ | $-0.2 \pm 0.0$ | $79.9 \pm 0.3$ | 80.0 | $62.3 \pm 0.6$ | 62.6 | $-0.0 \pm 0.0$ | $79.8 \pm 0.5$ | 79.8 | $62.0 \pm 0.5$ | 62.0 |
| GCN | 0.1 | $5.4 \pm 1.0$ | $80.8 \pm 1.1$ | 75.4 | $94.8 \pm 2.4$ | 83.9 | $4.9 \pm 1.0$ | $81.3 \pm 1.1$ | 76.4 | $95.0 \pm 1.7$ | 88.8 |
|  | 1.0 | $4.0 \pm 1.0$ | $81.0 \pm 0.5$ | 76.9 | $95.1 \pm 1.3$ | 88.4 | $1.3 \pm 0.0$ | $81.8 \pm 0.4$ | 80.4 | $95.5 \pm 0.6$ | 93.7 |
|  | 2.0 | $1.4 \pm 0.0$ | $81.2 \pm 0.4$ | 79.8 | $95.6 \pm 0.4$ | 92.6 | $0.9 \pm 0.0$ | $82.0 \pm 0.4$ | 81.1 | $95.8 \pm 0.4$ | 95.2 |
|  | 8.0 | $0.4 \pm 0.0$ | $81.3 \pm 0.2$ | 80.9 | $95.7 \pm 0.3$ | 95.9 | $-0.4 \pm 0.0$ | $81.6 \pm 0.3$ | 81.9 | $95.9 \pm 0.3$ | 95.9 |
|  | $\infty$ | $0.0 \pm 0.0$ | $81.7 \pm 0.0$ | 81.7 | $95.7 \pm 0.0$ | 95.7 | $0.0 \pm 0.0$ | $81.8 \pm 0.0$ | 81.8 | $96.2 \pm 0.0$ | 96.2 |
| GConv | 0.1 | $5.5 \pm 1.0$ | $76.5 \pm 1.0$ | 71.0 | $54.6 \pm 0.3$ | 56.6 | $1.4 \pm 1.0$ | $75.6 \pm 1.1$ | 74.1 | $54.3 \pm 0.3$ | 57.0 |
|  | 1.0 | $4.6 \pm 1.0$ | $75.8 \pm 0.5$ | 71.2 | $54.3 \pm 0.4$ | 56.7 | $1.0 \pm 1.0$ | $75.2 \pm 0.7$ | 74.3 | $54.6 \pm 0.6$ | 54.9 |
|  | 2.0 | $0.7 \pm 1.0$ | $77.0 \pm 1.1$ | 76.4 | $54.3 \pm 0.3$ | 54.7 | $-1.4 \pm 1.0$ | $74.8 \pm 1.0$ | 76.2 | $54.6 \pm 0.5$ | 54.0 |
|  | 8.0 | $1.1 \pm 0.0$ | $76.4 \pm 0.4$ | 75.3 | $54.2 \pm 0.4$ | 54.5 | $-0.8 \pm 0.0$ | $74.8 \pm 0.5$ | 75.6 | $54.1 \pm 0.3$ | 53.6 |
|  | $\infty$ | $0.0 \pm 0.0$ | $74.0 \pm 0.0$ | 74.0 | $53.9 \pm 0.0$ | 53.9 | $0.0 \pm 0.0$ | $74.0 \pm 0.0$ | 74.0 | $53.9 \pm 0.0$ | 53.9 |
| GT | 0.1 | $7.5 \pm 0.0$ | $79.5 \pm 0.5$ | 72.1 | $62.4 \pm 0.6$ | 58.3 | $4.8 \pm 0.0$ | $79.4 \pm 0.4$ | 74.6 | $62.2 \pm 0.5$ | 60.4 |
|  | 1.0 | $5.3 \pm 0.0$ | $79.6 \pm 0.5$ | 74.3 | $62.3 \pm 0.6$ | 59.1 | $1.0 \pm 0.0$ | $79.2 \pm 0.5$ | 78.2 | $62.1 \pm 0.4$ | 61.0 |
|  | 2.0 | $2.7 \pm 0.0$ | $79.6 \pm 0.4$ | 76.9 | $62.1 \pm 0.4$ | 60.8 | $0.4 \pm 0.0$ | $79.4 \pm 0.3$ | 79.0 | $62.6 \pm 0.6$ | 62.0 |
|  | 8.0 | $-0.6 \pm 0.0$ | $79.7 \pm 0.5$ | 80.2 | $62.4 \pm 0.5$ | 62.6 | $0.1 \pm 0.0$ | $79.7 \pm 0.4$ | 79.6 | $62.5 \pm 0.6$ | 62.5 |
|  | $\infty$ | $-0.4 \pm 1.0$ | $79.4 \pm 0.6$ | 79.7 | $62.4 \pm 0.5$ | 61.9 | $0.8 \pm 1.0$ | $79.5 \pm 0.6$ | 78.7 | $62.2 \pm 0.5$ | 62.5 |
| SAGE | 0.1 | $5.3 \pm 1.0$ | $80.8 \pm 1.1$ | 75.5 | $79.9 \pm 2.1$ | 72.6 | $4.5 \pm 1.0$ | $81.3 \pm 1.1$ | 76.8 | $78.6 \pm 1.2$ | 74.7 |
|  | 1.0 | $3.4 \pm 1.0$ | $81.1 \pm 0.5$ | 77.7 | $79.5 \pm 0.5$ | 75.9 | $1.5 \pm 0.0$ | $81.5 \pm 0.4$ | 80.0 | $79.3 \pm 0.4$ | 78.1 |
|  | 2.0 | $1.8 \pm 0.0$ | $81.3 \pm 0.3$ | 79.5 | $79.7 \pm 0.6$ | 76.5 | $0.4 \pm 0.0$ | $81.6 \pm 0.2$ | 81.2 | $79.6 \pm 0.5$ | 78.5 |
|  | 8.0 | $0.4 \pm 0.0$ | $81.4 \pm 0.1$ | 81.1 | $80.0 \pm 0.4$ | 80.1 | $-0.1 \pm 0.0$ | $81.5 \pm 0.2$ | 81.7 | $79.6 \pm 0.5$ | 79.8 |
|  | $\infty$ | $0.0 \pm 0.0$ | $81.8 \pm 0.0$ | 81.8 | $79.3 \pm 0.0$ | 79.3 | $0.0 \pm 0.0$ | $81.8 \pm 0.0$ | 81.8 | $79.3 \pm 0.0$ | 79.3 |

Table 9: Results on LastFM.

| Model | $\epsilon$ | GP-t $\overline{\Delta_{\text{Acc}}}$ | $\text{Acc}_{\text{GP}}$ | $\text{Acc}_{\text{b}}$ | $\text{AUC}_{\text{GP}}$ | $\text{AUC}_{\text{b}}$ | GP-m $\overline{\Delta_{\text{Acc}}}$ | $\text{Acc}_{\text{GP}}$ | $\text{Acc}_{\text{b}}$ | $\text{AUC}_{\text{GP}}$ | $\text{AUC}_{\text{b}}$ |
|---|---|---|---|---|---|---|---|---|---|---|---|
| GAT | 0.1 | $7.4 \pm 1.0$ | $82.5 \pm 1.0$ | 75.1 | $57.9 \pm 1.0$ | 54.4 | $3.2 \pm 1.0$ | $83.1 \pm 0.8$ | 79.9 | $57.4 \pm 0.5$ | 56.7 |
|  | 1.0 | $2.8 \pm 1.0$ | $83.3 \pm 0.9$ | 80.5 | $57.6 \pm 0.5$ | 55.3 | $2.4 \pm 1.0$ | $82.8 \pm 0.8$ | 80.4 | $57.6 \pm 0.4$ | 57.0 |
|  | 2.0 | $1.2 \pm 1.0$ | $83.6 \pm 0.7$ | 82.3 | $57.7 \pm 0.5$ | 56.3 | $-0.8 \pm 1.0$ | $83.3 \pm 1.1$ | 84.1 | $57.4 \pm 0.6$ | 58.0 |
|  | 8.0 | $-1.9 \pm 1.0$ | $82.8 \pm 0.9$ | 84.7 | $57.8 \pm 0.4$ | 58.6 | $2.6 \pm 1.0$ | $84.1 \pm 0.8$ | 81.5 | $58.1 \pm 0.6$ | 56.9 |
|  | $\infty$ | $-0.5 \pm 1.0$ | $82.9 \pm 0.8$ | 83.5 | $57.5 \pm 0.3$ | 57.8 | $0.4 \pm 1.0$ | $83.5 \pm 0.6$ | 83.1 | $57.5 \pm 0.3$ | 57.6 |
| GATv2 | 0.1 | $4.9 \pm 2.0$ | $74.6 \pm 1.9$ | 69.8 | $54.8 \pm 0.5$ | 52.6 | $1.3 \pm 2.0$ | $77.0 \pm 1.8$ | 75.7 | $55.1 \pm 0.4$ | 54.8 |
|  | 1.0 | $6.2 \pm 2.0$ | $77.1 \pm 1.9$ | 70.9 | $55.1 \pm 0.3$ | 52.7 | $-1.6 \pm 2.0$ | $75.6 \pm 1.5$ | 77.2 | $55.0 \pm 0.5$ | 54.8 |
|  | 2.0 | $6.0 \pm 4.0$ | $77.9 \pm 3.5$ | 71.9 | $54.9 \pm 0.4$ | 53.2 | $0.4 \pm 1.0$ | $76.9 \pm 1.5$ | 76.5 | $55.1 \pm 0.4$ | 55.1 |
|  | 8.0 | $-1.7 \pm 2.0$ | $75.6 \pm 1.8$ | 77.3 | $54.8 \pm 0.5$ | 54.6 | $-1.7 \pm 1.0$ | $76.9 \pm 1.1$ | 78.6 | $55.2 \pm 0.3$ | 54.6 |
|  | $\infty$ | $0.2 \pm 0.0$ | $77.1 \pm 0.4$ | 76.9 | $55.5 \pm 0.2$ | 55.7 | $-0.2 \pm 0.0$ | $77.1 \pm 0.4$ | 77.3 | $55.5 \pm 0.1$ | 55.3 |
| GCN | 0.1 | $3.9 \pm 0.0$ | $86.5 \pm 0.4$ | 82.6 | $93.5 \pm 2.0$ | 80.8 | $3.2 \pm 0.0$ | $86.3 \pm 0.5$ | 83.1 | $94.2 \pm 1.2$ | 87.4 |
|  | 1.0 | $2.3 \pm 0.0$ | $86.5 \pm 0.2$ | 84.2 | $94.0 \pm 1.2$ | 84.7 | $1.2 \pm 0.0$ | $86.5 \pm 0.2$ | 85.3 | $94.9 \pm 0.6$ | 90.9 |
|  | 2.0 | $2.0 \pm 0.0$ | $86.5 \pm 0.2$ | 84.5 | $94.6 \pm 0.8$ | 89.2 | $0.3 \pm 0.0$ | $86.6 \pm 0.1$ | 86.2 | $94.5 \pm 0.5$ | 94.5 |
|  | 8.0 | $0.1 \pm 0.0$ | $86.6 \pm 0.1$ | 86.4 | $94.7 \pm 0.5$ | 94.9 | $-0.2 \pm 0.0$ | $86.5 \pm 0.1$ | 86.7 | $94.0 \pm 0.4$ | 93.5 |
|  | $\infty$ | $0.0 \pm 0.0$ | $86.5 \pm 0.0$ | 86.5 | $94.8 \pm 0.0$ | 94.8 | $0.0 \pm 0.0$ | $86.6 \pm 0.0$ | 86.6 | $94.5 \pm 0.0$ | 94.5 |
| GConv | 0.1 | $4.0 \pm 1.0$ | $70.5 \pm 1.2$ | 66.5 | $51.9 \pm 0.2$ | 53.3 | $1.1 \pm 3.0$ | $70.4 \pm 3.2$ | 69.3 | $52.5 \pm 0.6$ | 53.7 |
|  | 1.0 | $2.2 \pm 1.0$ | $71.6 \pm 1.3$ | 69.4 | $52.2 \pm 0.3$ | 51.8 | $2.0 \pm 2.0$ | $69.1 \pm 1.8$ | 67.1 | $52.7 \pm 0.8$ | 53.1 |
|  | 2.0 | $2.4 \pm 1.0$ | $69.9 \pm 1.3$ | 67.5 | $52.6 \pm 0.7$ | 52.9 | $-2.8 \pm 3.0$ | $70.0 \pm 2.5$ | 72.7 | $52.2 \pm 0.3$ | 52.8 |
|  | 8.0 | $1.3 \pm 2.0$ | $69.6 \pm 2.2$ | 68.4 | $52.5 \pm 0.6$ | 52.4 | $1.2 \pm 2.0$ | $69.7 \pm 2.4$ | 68.5 | $52.5 \pm 0.6$ | 52.6 |
|  | $\infty$ | $0.0 \pm 0.0$ | $66.6 \pm 0.0$ | 66.6 | $51.8 \pm 0.0$ | 51.8 | $0.0 \pm 0.0$ | $66.6 \pm 0.0$ | 66.6 | $51.8 \pm 0.0$ | 51.8 |
| GT | 0.1 | $8.3 \pm 1.0$ | $80.2 \pm 0.8$ | 71.9 | $58.8 \pm 0.5$ | 54.6 | $3.5 \pm 1.0$ | $80.3 \pm 0.7$ | 76.8 | $58.9 \pm 0.3$ | 57.5 |
|  | 1.0 | $6.8 \pm 1.0$ | $80.5 \pm 0.8$ | 73.7 | $59.4 \pm 0.4$ | 55.4 | $2.8 \pm 1.0$ | $81.0 \pm 0.6$ | 78.2 | $59.1 \pm 0.4$ | 57.9 |
|  | 2.0 | $5.7 \pm 1.0$ | $79.8 \pm 0.9$ | 74.2 | $59.4 \pm 0.4$ | 56.5 | $0.5 \pm 0.0$ | $80.6 \pm 0.5$ | 80.1 | $59.0 \pm 0.4$ | 58.2 |
|  | 8.0 | $-1.0 \pm 1.0$ | $80.1 \pm 0.8$ | 81.1 | $59.3 \pm 0.3$ | 59.0 | $-1.2 \pm 1.0$ | $79.9 \pm 0.9$ | 81.2 | $58.9 \pm 0.4$ | 58.8 |
|  | $\infty$ | $-0.4 \pm 0.0$ | $80.3 \pm 0.5$ | 80.7 | $58.9 \pm 0.2$ | 58.9 | $-0.4 \pm 1.0$ | $80.4 \pm 0.6$ | 80.8 | $58.8 \pm 0.2$ | 59.0 |
| SAGE | 0.1 | $4.6 \pm 1.0$ | $84.6 \pm 0.8$ | 80.0 | $85.9 \pm 1.2$ | 83.7 | $3.2 \pm 1.0$ | $84.9 \pm 0.9$ | 81.8 | $85.2 \pm 0.8$ | 83.2 |
|  | 1.0 | $5.7 \pm 1.0$ | $84.6 \pm 0.5$ | 78.9 | $86.0 \pm 1.1$ | 83.5 | $3.1 \pm 1.0$ | $85.4 \pm 0.7$ | 82.3 | $85.2 \pm 1.0$ | 84.7 |
|  | 2.0 | $1.8 \pm 0.0$ | $84.6 \pm 0.3$ | 82.7 | $85.7 \pm 0.8$ | 85.4 | $1.8 \pm 1.0$ | $85.1 \pm 0.8$ | 83.4 | $85.8 \pm 0.4$ | 85.4 |
|  | 8.0 | $-0.0 \pm 1.0$ | $85.3 \pm 0.6$ | 85.3 | $85.7 \pm 0.9$ | 83.5 | $1.2 \pm 0.0$ | $85.3 \pm 0.5$ | 84.2 | $85.8 \pm 0.6$ | 85.1 |
|  | $\infty$ | $0.0 \pm 0.0$ | $86.0 \pm 0.0$ | 86.0 | $86.1 \pm 0.0$ | 86.1 | $0.0 \pm 0.0$ | $86.0 \pm 0.0$ | 86.0 | $86.1 \pm 0.0$ | 86.1 |

Table 10: Results on PubMed.

| Model | $\epsilon$ | GP-t | | | | | GP-m | | | | |
|---|---|---|---|---|---|---|---|---|---|---|---|
| | | $\overline{\Delta_{\mathrm{Acc}}}$ | $\mathrm{Acc_{GP}}$ | $\mathrm{Acc_b}$ | $\mathrm{AUC_{GP}}$ | $\mathrm{AUC_b}$ | $\overline{\Delta_{\mathrm{Acc}}}$ | $\mathrm{Acc_{GP}}$ | $\mathrm{Acc_b}$ | $\mathrm{AUC_{GP}}$ | $\mathrm{AUC_b}$ |
| GAT | 0.1 | $1.9 \pm 0.0$ | $82.4 \pm 0.1$ | 80.5 | $71.6 \pm 1.1$ | 67.3 | $1.8 \pm 0.0$ | $82.4 \pm 0.2$ | 80.6 | $72.6 \pm 0.9$ | 72.0 |
| | 1.0 | $1.4 \pm 0.0$ | $82.4 \pm 0.1$ | 81.1 | $72.4 \pm 1.4$ | 65.4 | $0.7 \pm 0.0$ | $82.5 \pm 0.1$ | 81.8 | $72.2 \pm 1.3$ | 73.8 |
| | 2.0 | $0.8 \pm 0.0$ | $82.5 \pm 0.1$ | 81.7 | $71.8 \pm 1.3$ | 69.7 | $0.0 \pm 0.0$ | $82.4 \pm 0.1$ | 82.4 | $71.9 \pm 1.4$ | 69.5 |
| | 8.0 | $0.1 \pm 0.0$ | $82.5 \pm 0.1$ | 82.4 | $72.8 \pm 1.2$ | 70.9 | $0.1 \pm 0.0$ | $82.5 \pm 0.1$ | 82.4 | $72.7 \pm 1.3$ | 73.3 |
| | $\infty$ | $-0.0 \pm 0.0$ | $82.5 \pm 0.0$ | 82.5 | $73.0 \pm 0.2$ | 73.2 | $-0.0 \pm 0.0$ | $82.5 \pm 0.0$ | 82.5 | $73.0 \pm 0.3$ | 72.8 |
| GATv2 | 0.1 | $1.6 \pm 0.0$ | $82.5 \pm 0.1$ | 80.9 | $64.5 \pm 0.6$ | 60.3 | $1.8 \pm 0.0$ | $82.5 \pm 0.1$ | 80.8 | $65.5 \pm 0.8$ | 61.7 |
| | 1.0 | $1.1 \pm 0.0$ | $82.4 \pm 0.1$ | 81.3 | $65.2 \pm 0.9$ | 61.4 | $0.7 \pm 0.0$ | $82.5 \pm 0.1$ | 81.9 | $65.8 \pm 0.8$ | 63.7 |
| | 2.0 | $0.8 \pm 0.0$ | $82.5 \pm 0.1$ | 81.7 | $64.3 \pm 0.8$ | 62.8 | $0.4 \pm 0.0$ | $82.5 \pm 0.1$ | 82.1 | $64.9 \pm 1.4$ | 64.9 |
| | 8.0 | $0.0 \pm 0.0$ | $82.5 \pm 0.1$ | 82.4 | $65.2 \pm 1.2$ | 65.3 | $-0.0 \pm 0.0$ | $82.5 \pm 0.0$ | 82.6 | $65.2 \pm 1.0$ | 66.5 |
| | $\infty$ | $-0.1 \pm 0.0$ | $82.6 \pm 0.1$ | 82.7 | $66.2 \pm 0.2$ | 66.2 | $0.0 \pm 0.0$ | $82.6 \pm 0.0$ | 82.6 | $66.2 \pm 0.2$ | 66.2 |
| GCN | 0.1 | $1.8 \pm 0.0$ | $82.3 \pm 0.1$ | 80.6 | $98.5 \pm 1.0$ | 85.9 | $1.8 \pm 0.0$ | $82.3 \pm 0.1$ | 80.5 | $98.7 \pm 0.7$ | 93.4 |
| | 1.0 | $1.0 \pm 0.0$ | $82.4 \pm 0.1$ | 81.4 | $98.9 \pm 0.5$ | 89.9 | $0.7 \pm 0.0$ | $82.3 \pm 0.1$ | 81.6 | $99.1 \pm 0.2$ | 97.2 |
| | 2.0 | $0.5 \pm 0.0$ | $82.4 \pm 0.1$ | 81.9 | $99.1 \pm 0.3$ | 94.4 | $0.0 \pm 0.0$ | $82.4 \pm 0.1$ | 82.3 | $99.1 \pm 0.2$ | 98.9 |
| | 8.0 | $0.1 \pm 0.0$ | $82.4 \pm 0.0$ | 82.3 | $99.1 \pm 0.2$ | 99.1 | $0.1 \pm 0.0$ | $82.4 \pm 0.1$ | 82.3 | $99.1 \pm 0.2$ | 98.8 |
| | $\infty$ | $0.0 \pm 0.0$ | $82.4 \pm 0.0$ | 82.4 | $99.2 \pm 0.0$ | 99.2 | $0.0 \pm 0.0$ | $82.4 \pm 0.0$ | 82.4 | $99.2 \pm 0.0$ | 99.2 |
| GConv | 0.1 | $2.0 \pm 0.0$ | $81.7 \pm 0.1$ | 79.7 | $74.8 \pm 1.0$ | 71.4 | $1.5 \pm 0.0$ | $81.8 \pm 0.1$ | 80.3 | $74.4 \pm 0.8$ | 76.0 |
| | 1.0 | $1.1 \pm 0.0$ | $81.7 \pm 0.1$ | 80.6 | $74.8 \pm 0.6$ | 73.1 | $0.6 \pm 0.0$ | $81.9 \pm 0.1$ | 81.3 | $74.6 \pm 0.7$ | 75.8 |
| | 2.0 | $0.7 \pm 0.0$ | $81.7 \pm 0.1$ | 81.1 | $74.1 \pm 0.7$ | 77.5 | $0.6 \pm 0.0$ | $81.9 \pm 0.1$ | 81.3 | $74.5 \pm 1.0$ | 74.4 |
| | 8.0 | $0.2 \pm 0.0$ | $81.8 \pm 0.1$ | 81.6 | $74.4 \pm 0.5$ | 74.9 | $0.0 \pm 0.0$ | $81.8 \pm 0.2$ | 81.7 | $73.8 \pm 1.3$ | 74.5 |
| | $\infty$ | $0.0 \pm 0.0$ | $82.0 \pm 0.0$ | 82.0 | $75.0 \pm 0.0$ | 75.0 | $0.0 \pm 0.0$ | $82.0 \pm 0.0$ | 82.0 | $75.0 \pm 0.0$ | 75.0 |
| GT | 0.1 | $1.9 \pm 0.0$ | $82.1 \pm 0.1$ | 80.1 | $64.3 \pm 0.2$ | 60.2 | $2.3 \pm 0.0$ | $82.0 \pm 0.1$ | 79.8 | $64.3 \pm 0.3$ | 63.2 |
| | 1.0 | $1.2 \pm 0.0$ | $82.0 \pm 0.1$ | 80.8 | $64.1 \pm 0.5$ | 61.0 | $0.8 \pm 0.0$ | $82.0 \pm 0.1$ | 81.2 | $64.1 \pm 0.2$ | 64.1 |
| | 2.0 | $1.0 \pm 0.0$ | $82.1 \pm 0.1$ | 81.0 | $64.3 \pm 0.2$ | 62.2 | $0.4 \pm 0.0$ | $82.0 \pm 0.1$ | 81.7 | $64.4 \pm 0.3$ | 64.4 |
| | 8.0 | $0.1 \pm 0.0$ | $82.1 \pm 0.1$ | 82.0 | $64.2 \pm 0.2$ | 64.1 | $0.3 \pm 0.0$ | $82.1 \pm 0.0$ | 81.9 | $64.4 \pm 0.3$ | 64.5 |
| | $\infty$ | $0.0 \pm 0.0$ | $82.1 \pm 0.0$ | 82.1 | $64.2 \pm 0.2$ | 64.3 | $-0.0 \pm 0.0$ | $82.1 \pm 0.0$ | 82.1 | $64.2 \pm 0.2$ | 64.0 |
| SAGE | 0.1 | $1.6 \pm 0.0$ | $82.0 \pm 0.2$ | 80.4 | $56.2 \pm 0.9$ | 56.1 | $1.9 \pm 0.0$ | $82.0 \pm 0.1$ | 80.1 | $55.9 \pm 0.5$ | 57.0 |
| | 1.0 | $1.1 \pm 0.0$ | $82.0 \pm 0.1$ | 80.9 | $55.6 \pm 0.6$ | 57.0 | $0.6 \pm 0.0$ | $82.0 \pm 0.1$ | 81.4 | $56.1 \pm 0.7$ | 56.6 |
| | 2.0 | $0.5 \pm 0.0$ | $82.0 \pm 0.1$ | 81.5 | $55.8 \pm 0.5$ | 54.8 | $0.2 \pm 0.0$ | $82.0 \pm 0.1$ | 81.8 | $55.9 \pm 0.6$ | 54.6 |
| | 8.0 | $-0.0 \pm 0.0$ | $82.0 \pm 0.1$ | 82.1 | $55.9 \pm 0.5$ | 55.7 | $0.1 \pm 0.0$ | $82.0 \pm 0.1$ | 82.0 | $56.0 \pm 0.6$ | 57.0 |
| | $\infty$ | $0.0 \pm 0.0$ | $82.0 \pm 0.0$ | 82.0 | $56.6 \pm 0.0$ | 56.6 | $0.0 \pm 0.0$ | $82.0 \pm 0.0$ | 82.0 | $56.6 \pm 0.0$ | 56.6 |

Table 11: Results on Facebook.

| Model | $\epsilon$ | $\overline{\Delta_{\text{Acc}}}$ | $\text{Acc}_{\text{GP}}$ | $\text{Acc}_{\text{b}}$ | $\text{AUC}_{\text{GP}}$ | $\text{AUC}_{\text{b}}$ | $\overline{\Delta_{\text{Acc}}}$ | $\text{Acc}_{\text{GP}}$ | $\text{Acc}_{\text{b}}$ | $\text{AUC}_{\text{GP}}$ | $\text{AUC}_{\text{b}}$ |
|---|---|---|---|---|---|---|---|---|---|---|---|
| | | | | GP-t | | | | | GP-m | | |
| GAT | 0.1 | $4.8 \pm 1.0$ | $90.5 \pm 1.1$ | 85.8 | $63.8 \pm 1.3$ | 60.8 | $3.1 \pm 0.0$ | $91.0 \pm 0.4$ | 87.9 | $64.8 \pm 0.6$ | 64.0 |
| | 1.0 | $4.1 \pm 1.0$ | $90.8 \pm 0.8$ | 86.6 | $65.2 \pm 1.7$ | 62.9 | $1.0 \pm 0.0$ | $91.0 \pm 0.2$ | 90.0 | $64.7 \pm 1.1$ | 66.1 |
| | 2.0 | $2.5 \pm 0.0$ | $90.8 \pm 0.4$ | 88.4 | $64.1 \pm 1.4$ | 64.9 | $0.2 \pm 0.0$ | $90.9 \pm 0.1$ | 90.7 | $64.4 \pm 1.0$ | 64.9 |
| | 8.0 | $0.1 \pm 0.0$ | $91.1 \pm 0.2$ | 91.0 | $63.8 \pm 1.1$ | 67.2 | $-0.3 \pm 0.0$ | $91.1 \pm 0.2$ | 91.4 | $64.9 \pm 1.3$ | 64.4 |
| | $\infty$ | $0.1 \pm 0.0$ | $90.9 \pm 0.1$ | 90.8 | $64.7 \pm 0.3$ | 64.8 | $-0.1 \pm 0.0$ | $90.9 \pm 0.1$ | 91.0 | $64.6 \pm 0.3$ | 64.3 |
| GATv2 | 0.1 | $4.9 \pm 1.0$ | $88.2 \pm 0.7$ | 83.3 | $58.5 \pm 1.8$ | 54.8 | $3.8 \pm 0.0$ | $88.6 \pm 0.4$ | 84.8 | $60.0 \pm 1.3$ | 57.0 |
| | 1.0 | $4.0 \pm 0.0$ | $88.1 \pm 0.4$ | 84.1 | $58.4 \pm 1.0$ | 54.8 | $1.0 \pm 1.0$ | $87.9 \pm 0.6$ | 86.9 | $59.3 \pm 1.4$ | 57.6 |
| | 2.0 | $3.2 \pm 0.0$ | $88.4 \pm 0.4$ | 85.1 | $58.8 \pm 1.0$ | 55.9 | $0.2 \pm 1.0$ | $88.3 \pm 0.5$ | 88.1 | $59.9 \pm 1.6$ | 59.3 |
| | 8.0 | $0.1 \pm 0.0$ | $88.6 \pm 0.5$ | 88.5 | $61.2 \pm 2.8$ | 59.7 | $-0.2 \pm 0.0$ | $88.2 \pm 0.4$ | 88.4 | $58.6 \pm 2.1$ | 57.8 |
| | $\infty$ | $0.1 \pm 0.0$ | $88.6 \pm 0.1$ | 88.4 | $61.7 \pm 0.4$ | 61.8 | $0.2 \pm 0.0$ | $88.6 \pm 0.1$ | 88.4 | $61.9 \pm 0.4$ | 61.3 |
| GCN | 0.1 | $4.8 \pm 1.0$ | $91.4 \pm 1.0$ | 86.6 | $98.1 \pm 1.5$ | 90.8 | $2.9 \pm 0.0$ | $91.8 \pm 0.3$ | 89.0 | $98.9 \pm 0.2$ | 96.9 |
| | 1.0 | $4.0 \pm 1.0$ | $91.6 \pm 0.7$ | 87.6 | $98.5 \pm 1.0$ | 93.2 | $1.1 \pm 0.0$ | $91.9 \pm 0.1$ | 90.8 | $99.0 \pm 0.1$ | 98.7 |
| | 2.0 | $3.0 \pm 0.0$ | $91.7 \pm 0.5$ | 88.7 | $98.8 \pm 0.4$ | 95.8 | $0.5 \pm 0.0$ | $91.9 \pm 0.1$ | 91.5 | $99.0 \pm 0.0$ | 99.1 |
| | 8.0 | $0.1 \pm 0.0$ | $91.9 \pm 0.0$ | 91.9 | $99.1 \pm 0.2$ | 99.2 | $0.1 \pm 0.0$ | $92.0 \pm 0.0$ | 91.9 | $99.0 \pm 0.1$ | 99.0 |
| | $\infty$ | $0.0 \pm 0.0$ | $91.9 \pm 0.0$ | 91.9 | $99.1 \pm 0.0$ | 99.1 | $0.0 \pm 0.0$ | $91.9 \pm 0.0$ | 91.9 | $99.1 \pm 0.0$ | 99.1 |
| GConv | 0.1 | $4.4 \pm 0.0$ | $88.5 \pm 0.3$ | 84.1 | $51.2 \pm 0.3$ | 52.4 | $2.5 \pm 0.0$ | $88.6 \pm 0.2$ | 86.1 | $51.2 \pm 0.2$ | 51.8 |
| | 1.0 | $4.0 \pm 0.0$ | $88.5 \pm 0.2$ | 84.6 | $51.1 \pm 0.2$ | 52.1 | $0.4 \pm 0.0$ | $88.5 \pm 0.2$ | 88.1 | $51.1 \pm 0.1$ | 51.7 |
| | 2.0 | $2.0 \pm 0.0$ | $88.5 \pm 0.3$ | 86.5 | $51.1 \pm 0.2$ | 52.2 | $0.1 \pm 0.0$ | $88.3 \pm 0.2$ | 88.2 | $51.2 \pm 0.4$ | 51.3 |
| | 8.0 | $-0.0 \pm 0.0$ | $88.5 \pm 0.3$ | 88.6 | $51.0 \pm 0.1$ | 51.1 | $0.5 \pm 0.0$ | $88.4 \pm 0.3$ | 88.0 | $51.1 \pm 0.1$ | 51.0 |
| | $\infty$ | $0.0 \pm 0.0$ | $88.5 \pm 0.0$ | 88.5 | $51.1 \pm 0.0$ | 51.1 | $0.0 \pm 0.0$ | $88.5 \pm 0.0$ | 88.5 | $51.1 \pm 0.0$ | 51.1 |
| GT | 0.1 | $4.7 \pm 0.0$ | $91.4 \pm 0.4$ | 86.7 | $60.1 \pm 0.8$ | 54.2 | $3.3 \pm 0.0$ | $91.5 \pm 0.2$ | 88.2 | $60.4 \pm 0.4$ | 57.2 |
| | 1.0 | $3.7 \pm 0.0$ | $91.5 \pm 0.2$ | 87.8 | $60.4 \pm 0.5$ | 55.7 | $1.0 \pm 0.0$ | $91.6 \pm 0.1$ | 90.6 | $60.3 \pm 0.4$ | 58.8 |
| | 2.0 | $2.6 \pm 0.0$ | $91.5 \pm 0.1$ | 88.9 | $60.2 \pm 0.6$ | 56.7 | $0.5 \pm 0.0$ | $91.6 \pm 0.1$ | 91.1 | $60.7 \pm 0.3$ | 59.5 |
| | 8.0 | $0.0 \pm 0.0$ | $91.6 \pm 0.1$ | 91.6 | $60.7 \pm 0.2$ | 60.0 | $0.1 \pm 0.0$ | $91.6 \pm 0.1$ | 91.4 | $60.5 \pm 0.6$ | 59.7 |
| | $\infty$ | $-0.0 \pm 0.0$ | $91.6 \pm 0.1$ | 91.6 | $60.3 \pm 0.4$ | 60.1 | $0.1 \pm 0.0$ | $91.6 \pm 0.1$ | 91.5 | $60.3 \pm 0.2$ | 60.1 |
| SAGE | 0.1 | $4.4 \pm 1.0$ | $91.3 \pm 0.9$ | 86.9 | $75.0 \pm 1.5$ | 69.8 | $2.6 \pm 1.0$ | $91.5 \pm 0.6$ | 88.9 | $75.0 \pm 1.4$ | 72.9 |
| | 1.0 | $3.6 \pm 1.0$ | $91.5 \pm 0.7$ | 87.8 | $74.7 \pm 0.7$ | 70.9 | $1.0 \pm 0.0$ | $91.7 \pm 0.2$ | 90.7 | $74.7 \pm 0.4$ | 76.0 |
| | 2.0 | $2.5 \pm 0.0$ | $91.6 \pm 0.4$ | 89.1 | $74.6 \pm 0.8$ | 71.0 | $0.4 \pm 0.0$ | $91.8 \pm 0.1$ | 91.4 | $75.2 \pm 0.5$ | 75.2 |
| | 8.0 | $-0.0 \pm 0.0$ | $91.8 \pm 0.0$ | 91.8 | $75.6 \pm 0.5$ | 76.2 | $0.1 \pm 0.0$ | $91.8 \pm 0.0$ | 91.7 | $75.8 \pm 0.9$ | 74.2 |
| | $\infty$ | $0.0 \pm 0.0$ | $91.9 \pm 0.0$ | 91.9 | $74.7 \pm 0.0$ | 74.7 | $0.0 \pm 0.0$ | $91.9 \pm 0.0$ | 91.9 | $74.7 \pm 0.0$ | 74.7 |

Table 12: Results on Amazon Photo.

| Model | $\epsilon$ | $\overline{\Delta_{\text{Acc}}}$ | $\text{Acc}_{\text{GP}}$ | $\text{Acc}_{\text{b}}$ | $\text{AUC}_{\text{GP}}$ | $\text{AUC}_{\text{b}}$ | $\overline{\Delta_{\text{Acc}}}$ | $\text{Acc}_{\text{GP}}$ | $\text{Acc}_{\text{b}}$ | $\text{AUC}_{\text{GP}}$ | $\text{AUC}_{\text{b}}$ |
|---|---|---|---|---|---|---|---|---|---|---|---|
| GAT | 0.1 | $13.9 \pm 2.0$ | $85.6 \pm 2.4$ | 71.8 | $80.9 \pm 3.9$ | 70.1 | $1.4 \pm 0.0$ | $86.3 \pm 0.5$ | 84.9 | $82.3 \pm 4.2$ | 79.4 |
|  | 1.0 | $15.4 \pm 2.0$ | $85.2 \pm 2.4$ | 69.8 | $80.5 \pm 4.6$ | 71.5 | $0.2 \pm 1.0$ | $86.2 \pm 1.1$ | 86.0 | $80.0 \pm 4.5$ | 75.9 |
|  | 2.0 | $12.8 \pm 2.0$ | $85.6 \pm 1.8$ | 72.8 | $83.5 \pm 1.9$ | 72.2 | $-0.2 \pm 1.0$ | $86.2 \pm 0.9$ | 86.3 | $82.4 \pm 3.6$ | 82.6 |
|  | 8.0 | $0.2 \pm 0.0$ | $86.4 \pm 0.3$ | 86.1 | $84.9 \pm 2.9$ | 73.6 | $-0.2 \pm 1.0$ | $86.3 \pm 1.1$ | 86.5 | $83.4 \pm 2.6$ | 83.1 |
|  | $\infty$ | $-0.0 \pm 0.0$ | $86.8 \pm 0.0$ | 86.8 | $83.9 \pm 0.5$ | 84.4 | $-0.0 \pm 0.0$ | $86.8 \pm 0.0$ | 86.8 | $83.7 \pm 0.6$ | 83.7 |
| GATv2 | 0.1 | $13.9 \pm 2.0$ | $83.8 \pm 2.4$ | 69.9 | $71.0 \pm 4.6$ | 61.7 | $2.6 \pm 2.0$ | $82.7 \pm 2.2$ | 80.1 | $70.2 \pm 4.2$ | 68.2 |
|  | 1.0 | $12.2 \pm 4.0$ | $82.6 \pm 4.0$ | 70.4 | $69.4 \pm 4.8$ | 57.2 | $-0.6 \pm 2.0$ | $83.7 \pm 1.9$ | 84.3 | $70.9 \pm 4.1$ | 72.4 |
|  | 2.0 | $13.9 \pm 5.0$ | $82.2 \pm 5.3$ | 68.3 | $68.3 \pm 2.6$ | 60.0 | $-0.9 \pm 1.0$ | $84.1 \pm 0.9$ | 85.0 | $72.0 \pm 3.2$ | 76.8 |
|  | 8.0 | $-1.2 \pm 1.0$ | $83.8 \pm 1.3$ | 85.0 | $72.9 \pm 3.4$ | 71.8 | $-1.1 \pm 2.0$ | $83.4 \pm 1.8$ | 84.5 | $72.6 \pm 3.4$ | 69.9 |
|  | $\infty$ | $0.0 \pm 0.0$ | $84.9 \pm 0.0$ | 84.9 | $76.7 \pm 0.8$ | 77.6 | $-0.0 \pm 0.0$ | $84.9 \pm 0.0$ | 84.9 | $76.6 \pm 0.9$ | 77.3 |
| GCN | 0.1 | $15.4 \pm 3.0$ | $84.1 \pm 2.9$ | 68.7 | $96.4 \pm 1.7$ | 89.2 | $0.5 \pm 2.0$ | $83.2 \pm 1.6$ | 82.7 | $96.2 \pm 0.9$ | 95.9 |
|  | 1.0 | $10.6 \pm 3.0$ | $83.7 \pm 3.5$ | 73.1 | $95.9 \pm 1.2$ | 92.3 | $-0.7 \pm 2.0$ | $83.5 \pm 1.9$ | 84.2 | $96.2 \pm 0.8$ | 95.8 |
|  | 2.0 | $12.8 \pm 3.0$ | $83.3 \pm 3.1$ | 70.4 | $95.3 \pm 0.6$ | 93.0 | $-3.8 \pm 3.0$ | $81.9 \pm 2.6$ | 85.7 | $96.1 \pm 1.0$ | 95.5 |
|  | 8.0 | $-1.7 \pm 2.0$ | $84.2 \pm 1.5$ | 85.9 | $95.7 \pm 0.9$ | 95.3 | $-4.4 \pm 2.0$ | $81.8 \pm 1.9$ | 86.2 | $96.0 \pm 0.6$ | 96.5 |
|  | $\infty$ | $0.0 \pm 0.0$ | $80.8 \pm 0.0$ | 80.8 | $96.3 \pm 0.0$ | 96.3 | $0.0 \pm 0.0$ | $80.8 \pm 0.0$ | 80.8 | $96.3 \pm 0.0$ | 96.3 |
| GConv | 0.1 | $-3.2 \pm 6.0$ | $64.1 \pm 6.1$ | 67.3 | $51.8 \pm 1.0$ | 52.0 | $-5.3 \pm 7.0$ | $70.8 \pm 7.1$ | 76.1 | $52.2 \pm 0.6$ | 51.7 |
|  | 1.0 | $5.4 \pm 5.0$ | $65.9 \pm 5.0$ | 60.5 | $52.3 \pm 0.9$ | 51.7 | $-7.7 \pm 9.0$ | $62.2 \pm 9.0$ | 69.9 | $52.1 \pm 0.7$ | 52.2 |
|  | 2.0 | $11.8 \pm 4.0$ | $66.3 \pm 4.3$ | 54.4 | $52.2 \pm 1.5$ | 51.8 | $-7.4 \pm 6.0$ | $63.0 \pm 6.2$ | 70.3 | $51.9 \pm 0.5$ | 51.8 |
|  | 8.0 | $-13.0 \pm 11.0$ | $60.8 \pm 10.5$ | 73.7 | $52.2 \pm 0.8$ | 52.8 | $-5.5 \pm 7.0$ | $65.5 \pm 6.6$ | 71.0 | $52.0 \pm 0.6$ | 51.7 |
|  | $\infty$ | $0.0 \pm 0.0$ | $61.3 \pm 0.0$ | 61.3 | $52.2 \pm 0.0$ | 52.2 | $0.0 \pm 0.0$ | $61.3 \pm 0.0$ | 61.3 | $52.2 \pm 0.0$ | 52.2 |
| GT | 0.1 | $8.7 \pm 1.0$ | $86.1 \pm 0.9$ | 77.3 | $81.5 \pm 6.6$ | 66.4 | $1.3 \pm 1.0$ | $86.4 \pm 0.6$ | 85.2 | $82.7 \pm 4.2$ | 74.5 |
|  | 1.0 | $7.6 \pm 1.0$ | $86.1 \pm 0.9$ | 78.6 | $78.4 \pm 5.5$ | 67.3 | $0.3 \pm 1.0$ | $86.4 \pm 0.7$ | 86.0 | $82.8 \pm 3.1$ | 73.8 |
|  | 2.0 | $3.4 \pm 1.0$ | $86.2 \pm 0.7$ | 82.8 | $80.8 \pm 4.2$ | 67.1 | $0.0 \pm 0.0$ | $86.4 \pm 0.5$ | 86.3 | $83.5 \pm 3.1$ | 77.1 |
|  | 8.0 | $0.0 \pm 0.0$ | $86.6 \pm 0.2$ | 86.6 | $82.5 \pm 2.0$ | 80.3 | $-0.2 \pm 0.0$ | $86.6 \pm 0.1$ | 86.8 | $84.7 \pm 2.4$ | 81.0 |
|  | $\infty$ | $-0.0 \pm 0.0$ | $86.7 \pm 0.0$ | 86.7 | $87.5 \pm 0.5$ | 87.6 | $0.1 \pm 0.0$ | $86.7 \pm 0.0$ | 86.6 | $87.3 \pm 0.6$ | 87.8 |
| SAGE | 0.1 | $14.9 \pm 2.0$ | $85.9 \pm 1.7$ | 71.0 | $85.1 \pm 6.9$ | 70.9 | $1.0 \pm 0.0$ | $86.3 \pm 0.3$ | 85.3 | $86.2 \pm 7.0$ | 79.0 |
|  | 1.0 | $14.6 \pm 1.0$ | $85.9 \pm 0.9$ | 71.3 | $78.4 \pm 2.1$ | 71.6 | $0.2 \pm 0.0$ | $86.3 \pm 0.2$ | 86.1 | $77.6 \pm 2.8$ | 83.0 |
|  | 2.0 | $13.0 \pm 2.0$ | $85.8 \pm 1.6$ | 72.7 | $80.5 \pm 4.3$ | 72.7 | $0.2 \pm 0.0$ | $86.4 \pm 0.2$ | 86.3 | $79.0 \pm 2.3$ | 83.0 |
|  | 8.0 | $-0.2 \pm 0.0$ | $86.7 \pm 0.1$ | 86.9 | $80.0 \pm 1.9$ | 80.0 | $0.1 \pm 0.0$ | $86.5 \pm 0.1$ | 86.4 | $78.7 \pm 2.0$ | 83.0 |
|  | $\infty$ | $0.0 \pm 0.0$ | $86.4 \pm 0.0$ | 86.4 | $93.8 \pm 0.0$ | 93.8 | $0.0 \pm 0.0$ | $86.4 \pm 0.0$ | 86.4 | $93.8 \pm 0.0$ | 93.8 |

### C.4 Additional results with GAT

We report results for $\epsilon = 0.1$ and different values of $\alpha$ and $\delta$, for all our experiments, as averaged values across 10 runs.

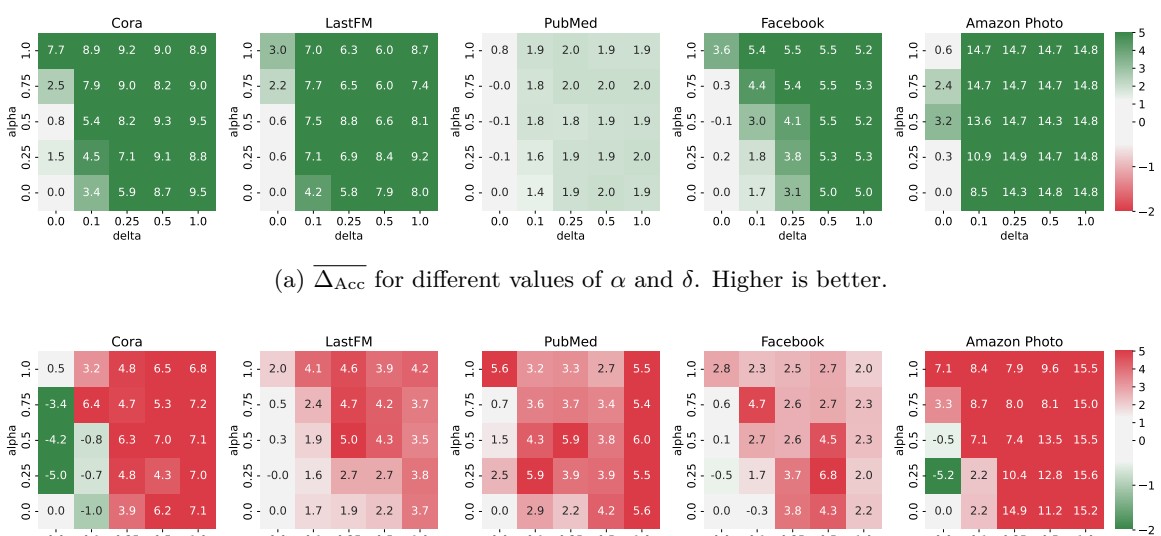

(a) $\overline{\Delta_{\text{Acc}}}$ for different values of $\alpha$ and $\delta$. Higher is better.

(b) $\overline{\Delta_{\text{AUC}}}$ for different values of $\alpha$ and $\delta$. Smaller is better.

Figure 4: Results for GP-t with GAT, for $\epsilon = 0.1$. Average values across 10 runs. The colormap is normalized to interpret all $\overline{\Delta_{\text{Acc}}} > 0$ and $\overline{\Delta_{\text{AUC}}} \leq 2$ as desirable.

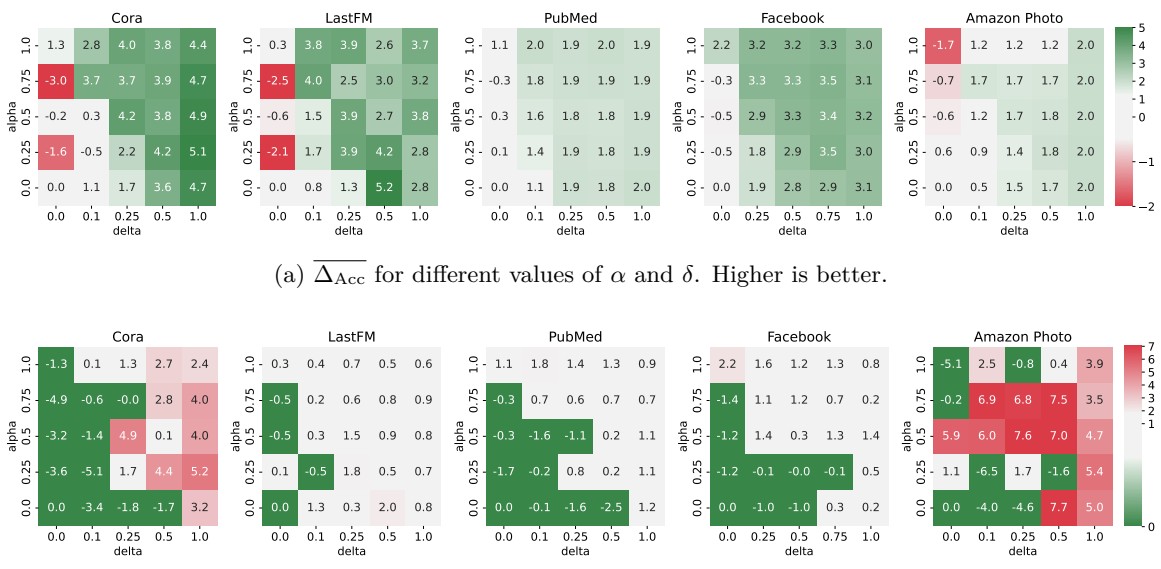

(a) $\overline{\Delta_{\text{Acc}}}$ for different values of $\alpha$ and $\delta$. Higher is better.

(b) $\overline{\Delta_{\text{AUC}}}$ for different values of $\alpha$ and $\delta$. Smaller is better.

Figure 5: Results for GP-m with GAT, for $\epsilon = 0.1$. Average values across 10 runs. The colormap is normalized to interpret all $\overline{\Delta_{\text{Acc}}} > 0$ and $\overline{\Delta_{\text{AUC}}} \leq 2$ as desirable.

## C.5   Additional results with SAGE

We report results for $\epsilon = 0.1$ and different values of $\alpha$ and $\delta$, for all our experiments, as averaged values across 10 runs.

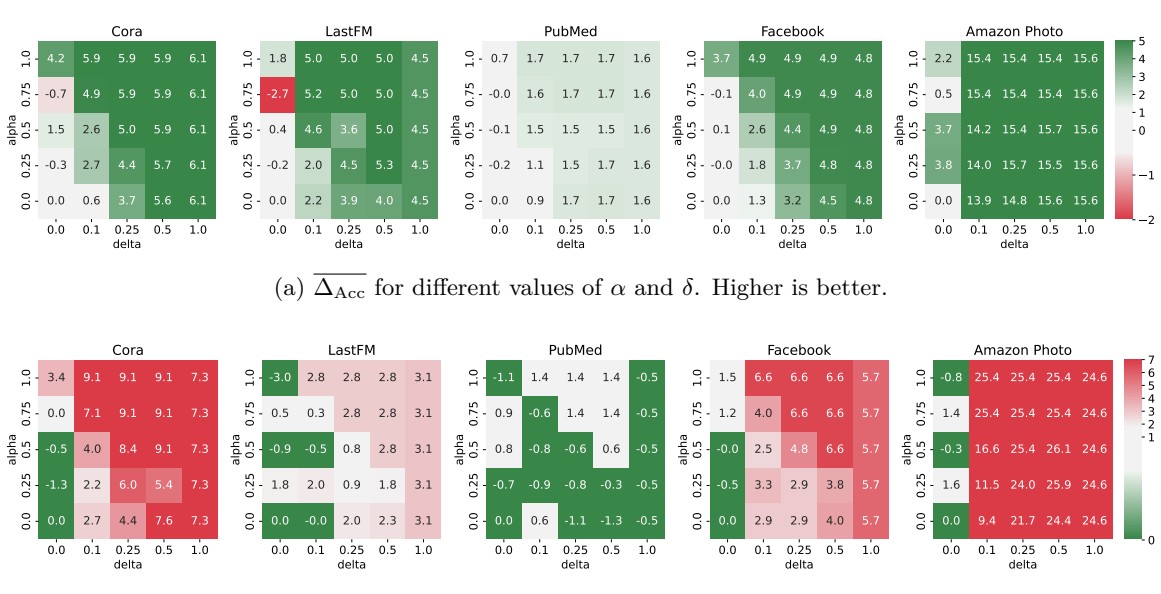

(a) $\overline{\Delta_{\mathrm{Acc}}}$ for different values of $\alpha$ and $\delta$. Higher is better.

(b) $\overline{\Delta_{\mathrm{AUC}}}$ for different values of $\alpha$ and $\delta$. Smaller is better.

Figure 6: Results for GP-t with SAGE, for $\epsilon = 0.1$. Average values across 10 runs. The colormap is normalized to interpret all $\overline{\Delta_{\mathrm{Acc}}} > 0$ and $\overline{\Delta_{\mathrm{AUC}}} \leq 2$ as desirable.

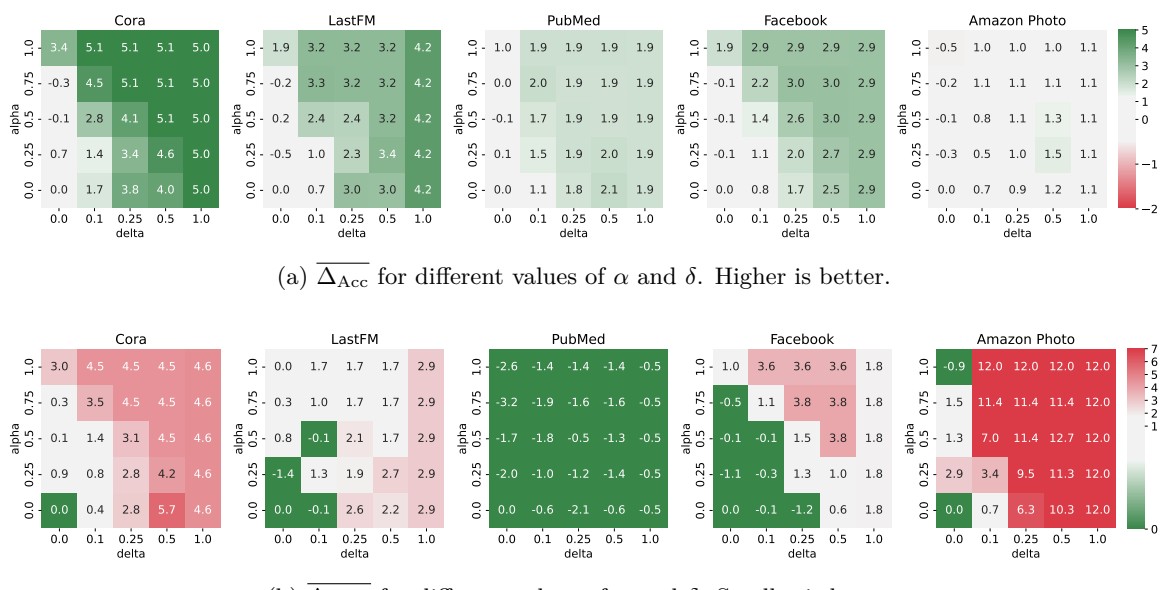

(a) $\overline{\Delta_{\mathrm{Acc}}}$ for different values of $\alpha$ and $\delta$. Higher is better.

(b) $\overline{\Delta_{\mathrm{AUC}}}$ for different values of $\alpha$ and $\delta$. Smaller is better.

Figure 7: Results for GP-m with SAGE, for $\epsilon = 0.1$. Average values across 10 runs. The colormap is normalized to interpret all $\overline{\Delta_{\mathrm{Acc}}} > 0$ and $\overline{\Delta_{\mathrm{AUC}}} \leq 2$ as desirable.

## C.6 Additional results with GCN

We report results for $\epsilon = 0.1$ and different values of $\alpha$ and $\delta$, for all our experiments, as averaged values across 10 runs.

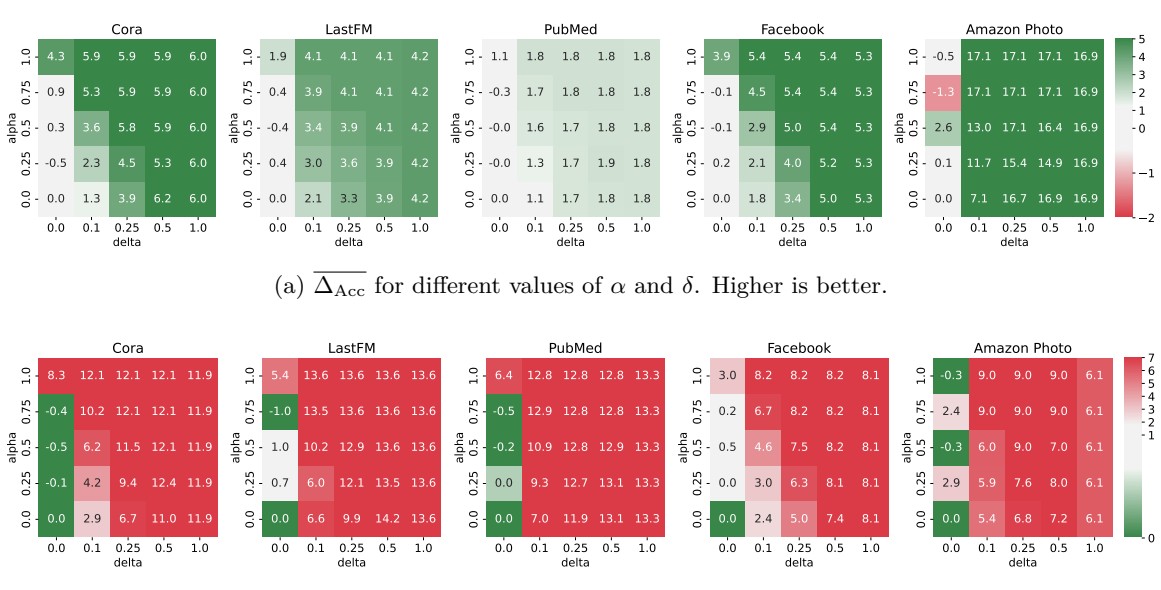

Figure 8: Results for GP-t with GCN, for $\epsilon = 0.1$. Average values across 10 runs. The colormap is normalized to interpret all $\overline{\Delta_{\text{Acc}}} > 0$ and $\overline{\Delta_{\text{AUC}}} \leq 2$ as desirable.

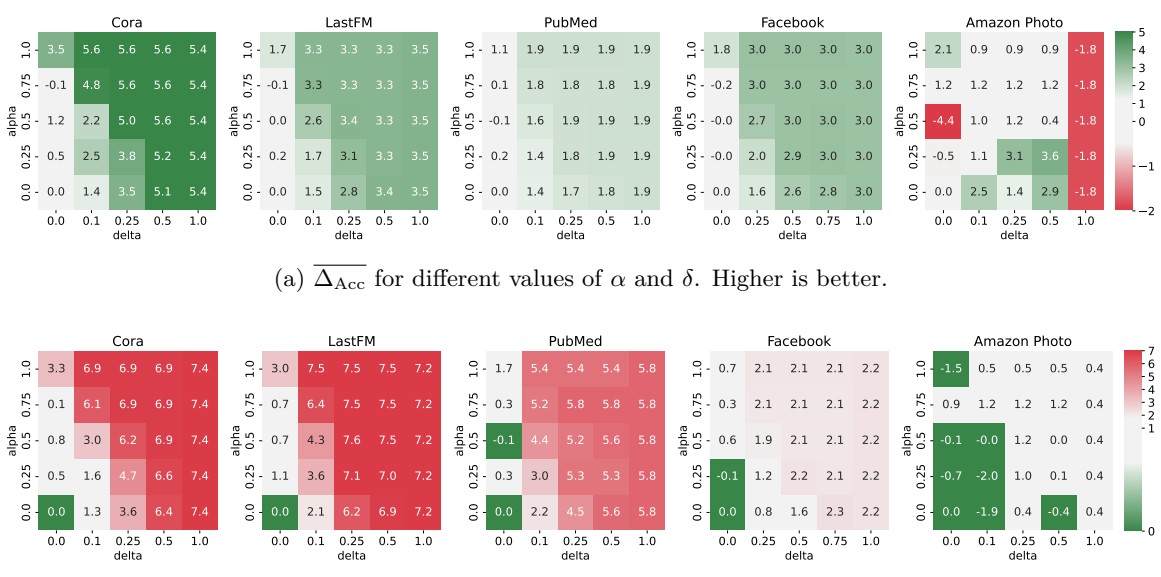

Figure 9: Results for GP-m with GCN, for $\epsilon = 0.1$. Average values across 10 runs. The colormap is normalized to interpret all $\overline{\Delta_{\text{Acc}}} > 0$ and $\overline{\Delta_{\text{AUC}}} \leq 2$ as desirable.

## C.7  Additional results with GATv2

We report results for $\epsilon = 0.1$ and different values of $\alpha$ and $\delta$, for all our experiments, as averaged values across 10 runs.

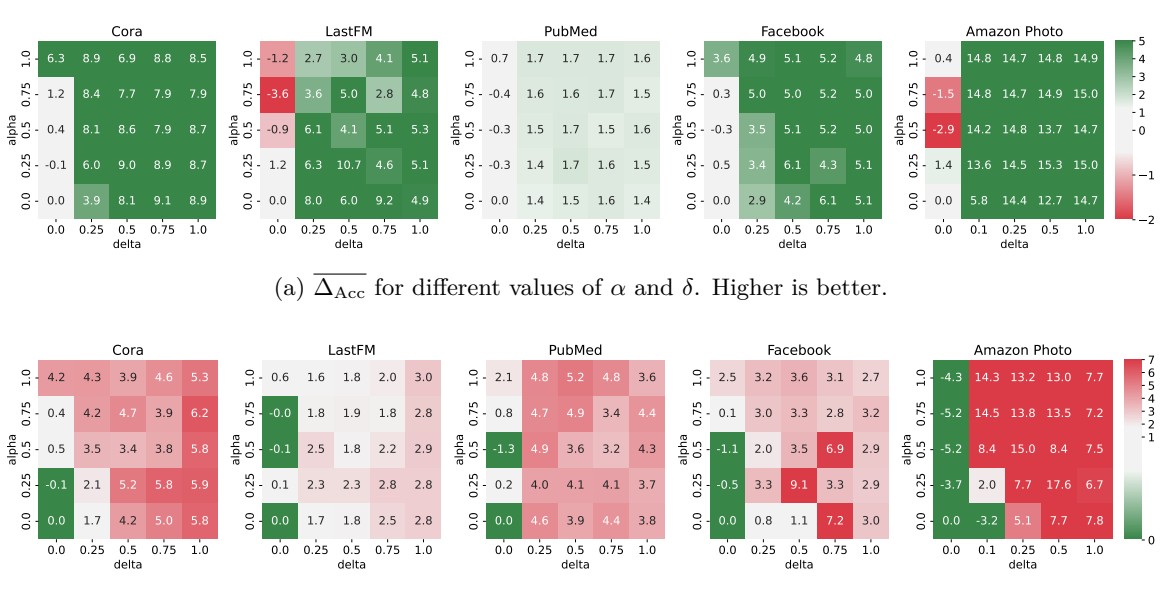

(a) $\overline{\Delta_{\mathrm{Acc}}}$ for different values of $\alpha$ and $\delta$. Higher is better.

(b) $\overline{\Delta_{\mathrm{AUC}}}$ for different values of $\alpha$ and $\delta$. Smaller is better.

Figure 10: Results for GP-t with GATv2, for $\epsilon = 0.1$. Average values across 10 runs. The colormap is normalized to interpret all $\overline{\Delta_{\mathrm{Acc}}} > 0$ and $\overline{\Delta_{\mathrm{AUC}}} \leq 2$ as desirable.

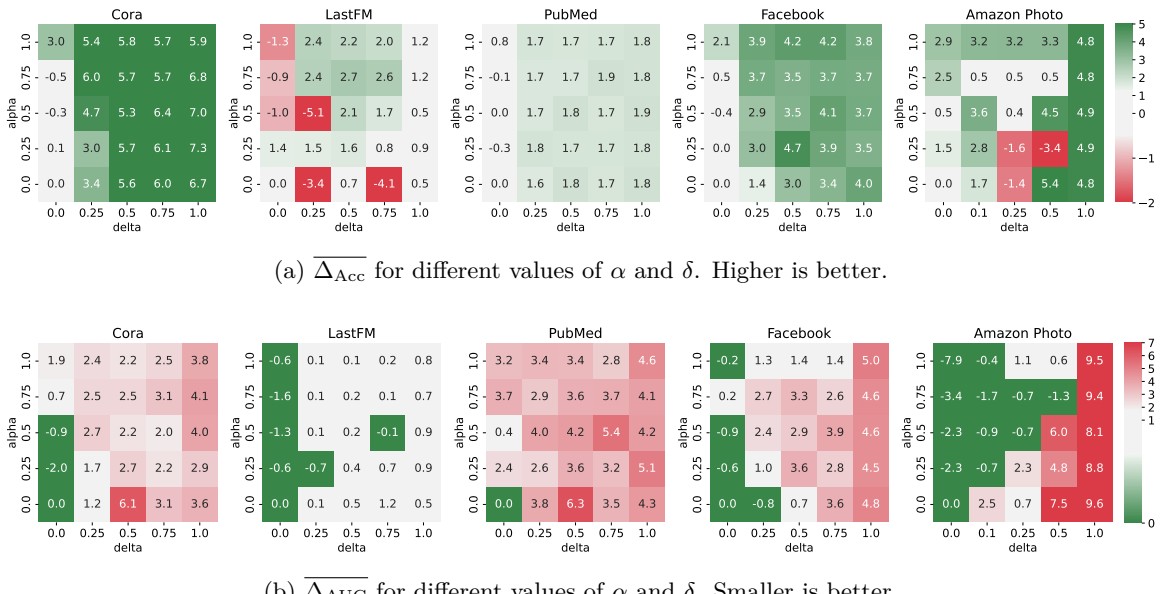

(a) $\overline{\Delta_{\mathrm{Acc}}}$ for different values of $\alpha$ and $\delta$. Higher is better.

(b) $\overline{\Delta_{\mathrm{AUC}}}$ for different values of $\alpha$ and $\delta$. Smaller is better.

Figure 11: Results for GP-m with GATv2, for $\epsilon = 0.1$. Average values across 10 runs. The colormap is normalized to interpret all $\overline{\Delta_{\mathrm{Acc}}} > 0$ and $\overline{\Delta_{\mathrm{AUC}}} \leq 2$ as desirable.

## C.8   Additional results with GT

We report results for $\epsilon = 0.1$ and different values of $\alpha$ and $\delta$, for all our experiments, as averaged values across 10 runs.

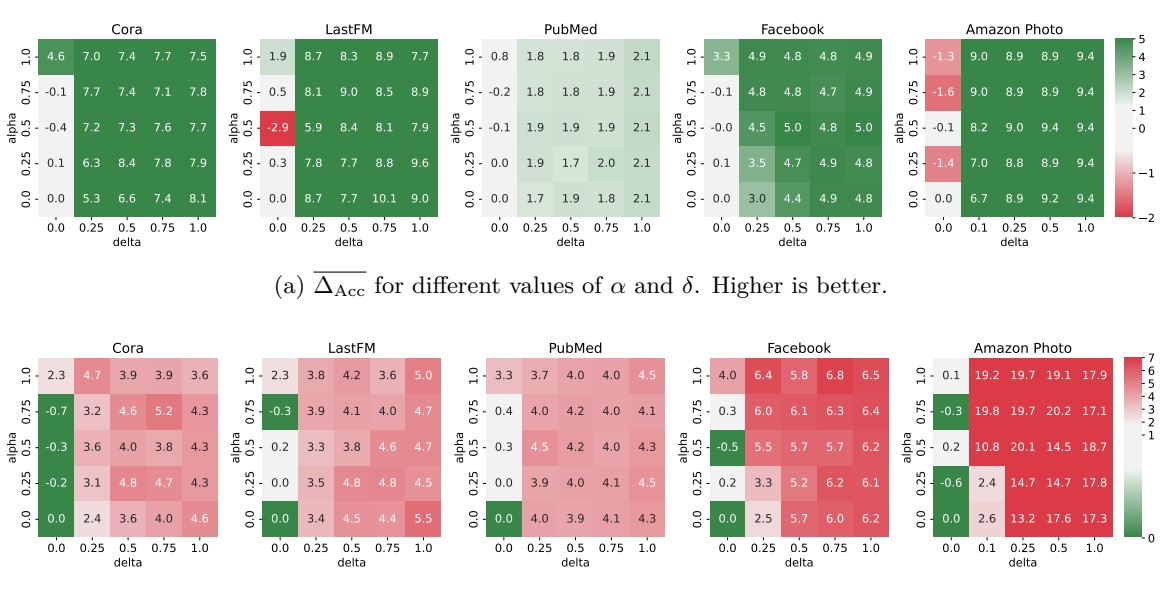

(a) $\overline{\Delta_{\mathrm{Acc}}}$ for different values of $\alpha$ and $\delta$. Higher is better.

(b) $\overline{\Delta_{\mathrm{AUC}}}$ for different values of $\alpha$ and $\delta$. Smaller is better.

Figure 12: Results for GP-t with GT, for $\epsilon = 0.1$. Average values across 10 runs. The colormap is normalized to interpret all $\overline{\Delta_{\mathrm{Acc}}} > 0$ and $\overline{\Delta_{\mathrm{AUC}}} \leq 2$ as desirable.

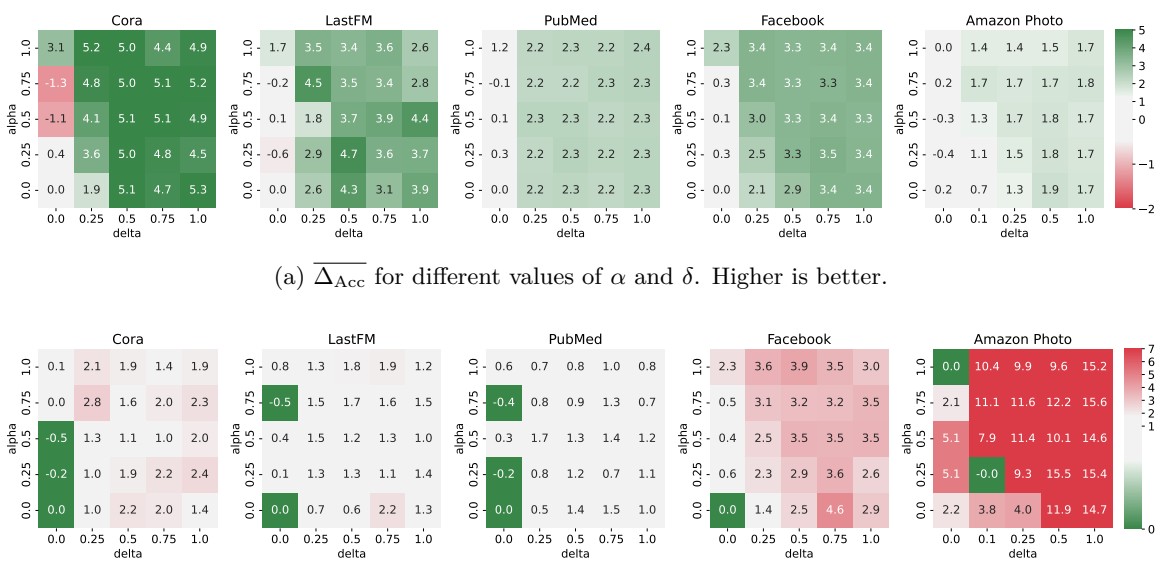

(a) $\overline{\Delta_{\mathrm{Acc}}}$ for different values of $\alpha$ and $\delta$. Higher is better.

(b) $\overline{\Delta_{\mathrm{AUC}}}$ for different values of $\alpha$ and $\delta$. Smaller is better.

Figure 13: Results for GP-m with GT, for $\epsilon = 0.1$. Average values across 10 runs. The colormap is normalized to interpret all $\overline{\Delta_{\mathrm{Acc}}} > 0$ and $\overline{\Delta_{\mathrm{AUC}}} \leq 2$ as desirable.

## C.9 Additional results with GConv

We report results for $\epsilon = 0.1$ and different values of $\alpha$ and $\delta$, for all our experiments, as averaged values across 10 runs.

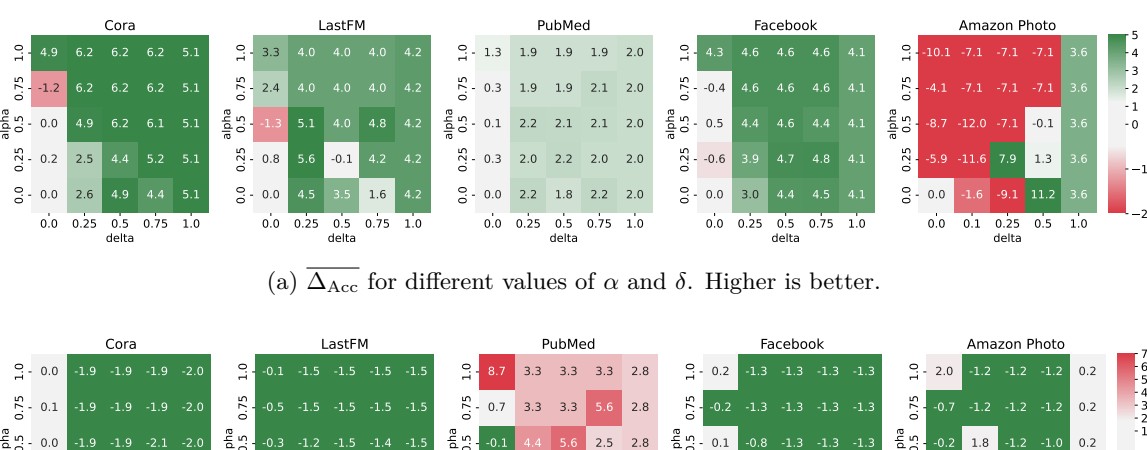

(a) $\overline{\Delta_{\text{Acc}}}$ for different values of $\alpha$ and $\delta$. Higher is better.

(b) $\overline{\Delta_{\text{AUC}}}$ for different values of $\alpha$ and $\delta$. Smaller is better.

Figure 14: Results for GP-t with GConv, for $\epsilon = 0.1$. Average values across 10 runs. The colormap is normalized to interpret all $\overline{\Delta_{\text{Acc}}} > 0$ and $\overline{\Delta_{\text{AUC}}} \leq 2$ as desirable.

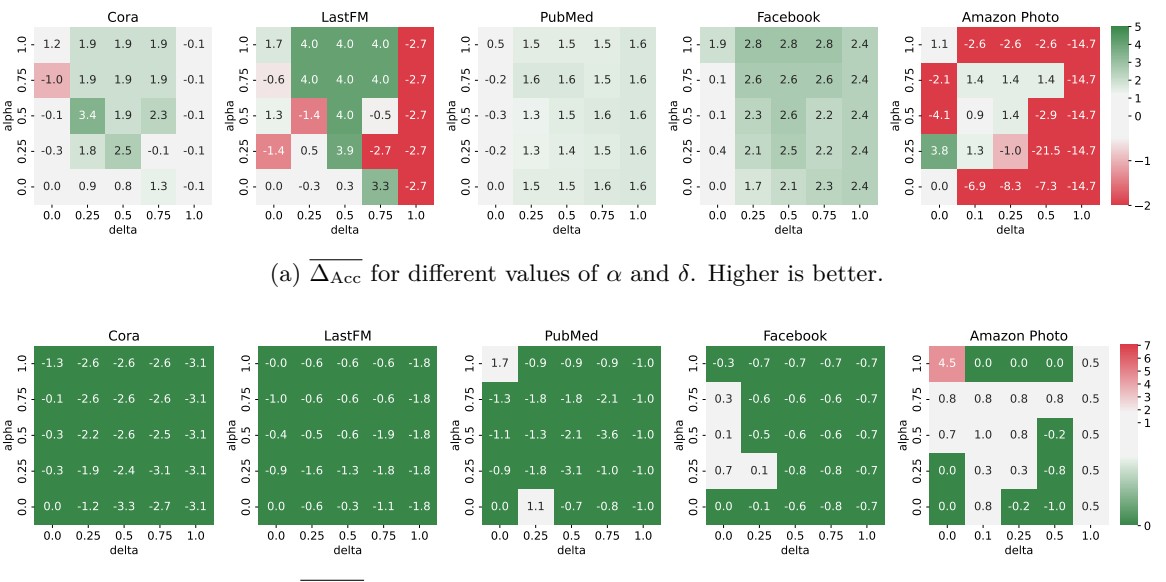

(a) $\overline{\Delta_{\text{Acc}}}$ for different values of $\alpha$ and $\delta$. Higher is better.

(b) $\overline{\Delta_{\text{AUC}}}$ for different values of $\alpha$ and $\delta$. Smaller is better.

Figure 15: Results for GP-m with GConv, for $\epsilon = 0.1$. Average values across 10 runs. The colormap is normalized to interpret all $\overline{\Delta_{\text{Acc}}} > 0$ and $\overline{\Delta_{\text{AUC}}} \leq 2$ as desirable.

