# OpenReview forum: "GraphPrivatizer: Improved Structural Differential Privacy for Graph Neural Networks"
_TMLR — Accepted by TMLR_

### Review · Reviewer_vXhW · 2024-06-18

**Summary Of Contributions:**

The paper investigates differentially private algorithms within the framework of graph neural networks, with a particular focus on edge-level differential privacy. The primary motivation is to address the limitations of existing methods, such as the Randomized Response technique, which can undesirably increase the connectivity of the underlying graph. The authors propose a novel algorithm that ensures $\epsilon$-edge set Local Differential Privacy (LDP). Empirical results demonstrate the relative effectiveness of the proposed approach.

**Audience:**

Yes

**Claims And Evidence:**

Yes

**Requested Changes:**

In addition to the previous weak points that should be addressed, a number of questions are also relevant and would be great to clarify in the manuscript:

- In the beginning of Section 4.2, you state: “new definition.” Can you clarify this statement? Specifically, how novel is your “introduced definition” of adjacent neighborhoods compared to other available node-level privacy definitions (such as [1])? Note that while you aim for edge-level privacy, you seem to be using a node-level definition (Definition 4.1).
- How can the similarity score be adapted in cases where nodes do not have labels? Generally, in the absence of node features, some crafted vectors (usually based on the node’s degree) are used. How can this be adapted in your case where the neighborhood is perturbed?
- In accordance with W2, it seems that the interaction between label and edge privacy is not considered in Theorem 4.4. Can you please clarify this possible non-dependence? Additionally, a more formal and complete proof of this algorithm would be beneficial.
- How does the algorithm scale with larger datasets? It would be great to provide results on larger datasets (perhaps OGBN-Arxiv?).
- Related to the previous question, how can the model be adapted for datasets that contain edge features? How can this information be used to compute similarities?
- In the experimental results, you use a 50:25:25 train/val/test split ratio (Appendix A). Is there a reason why you did not use the public folds available with some datasets (such as Cora and PubMed)?
- As a suggestion, it would be great to expand your results to the graph classification setting. I believe your algorithm can be easily adapted to this setting as well.

---
[1] Preserving Node-level Privacy in Graph Neural Networks, Xiang & Al. - 2024.

**Strengths And Weaknesses:**

**Strong points:**
- The problem considered here is very interesting and relevant.
- The algorithm is theoretically validated to achieve $\epsilon$-edge set LDP.
- The proposed algorithm seems novel and is "model-agnostic," making it easily applicable to the majority of available GNNs (also also shown in the experimental section).
- The empirical results look promising.

**Weak points:**
- [W1] In section 2, you claim: “individual nodes have only access to their own edges, features, and labels,” but in the algorithm, each node has access not only to its neighborhood but also to its 2-hop neighbors (and their labels to compute similarity). Can you please clarify this?
- [W2] The authors use a similarity score based on the cosine similarity of node features. I believe more advanced and relevant similarity scores could yield better results. Additionally, considering feature privacy can be confusing as the algorithm becomes a two-step randomization. Specifically, ensuring node features' privacy will likely affect edge-level privacy.
- [W3] The code implementation seems to be missing, raising possible questions about the validity and reproducibility of the empirical results. Could you please provide it as part of the submission?
- [W4] The paper lacks a discussion on time complexity. Specifically, how does the algorithm scale with larger datasets?
- [W5] On a minor personal note, the ordering of the sections was confusing. It would be better to divide all the concepts into a preliminary section and then adapt the problem setup section to include everything related to the authors’ work and hypotheses.

---

> ### Author Response · Authors · 2024-06-27
> **Reply to Reviewer vXhW**
>
> Thank you very much for your comments and detailed feedback, which we are happy to address.
>
> **W1**
>
>  In Section 2 we refer to the noise-free information that a node has access to. As specified in the problem statement, the only noise-free information a node can have access to is its own features, labels, and edges. We will make this explicit and more clear in a revised version of the submission. To further clarify, while for each node $n$ the perturbation is performed within the 2-hop neighborhood of $n$, $n$ _does not_ have access to noise-free labels or features of any node except itself. Similarities are in fact computed using noisy features. With reference to Algorithm 2, a node $n$ queries one of its neighbors $u$ for replacement candidates and it is $u$ which computes the similarity scores within its own 1-hop neighborhood. That is, $n$ is not aware of which nodes constitute the complete neighborhood of $u$.
>
> **W2**
>
> In preliminary experiments the cosine similarity was a natural choice to compute the similarity between embeddings and provided the most promising results. We are however happy to consider more advanced scores you may suggest. However, as the similarity is computed starting from noisy/perturbed feature vectors, it is difficult to estimate how beneficial more advanced scores may be. Moreover, our local privacy setting may make it difficult to use similarity metrics which can be learnt on data. In this setting, non-parametric, non-learnable similarity notions are the most natural choice if no prior information on data is available. To this end, we performed some additional experiments using the Euclidean distance to compute similarities, which led to worse results. In the following table we computed the difference between the results obtained with cosine similarity and the ones obtained with the l2 distance. Cosine similarity leads to better accuracy ($\overline{\Delta_{\text{Acc}}}>0$), with comparable privacy ($\overline{\Delta_{\text{AUC}}} \approx 0$).
>
> |          | $\overline{\Delta_{\text{Acc}}}$ | $\overline{\Delta_{\text{AUC}}}$ |
> |----------|:------------------------------:|:------------------------------:|
> |   Cora   | $4.8\pm 1.2    $                 | $2 \pm 3     $                   |
> | Facebook | $1.6\pm 0.5   $                  | $1 \pm 1   $                     |
> |  LastFM  | $2.4 \pm 1.4   $                 | $1 \pm 1   $                     |
> |   Photo  | $-0.9 \pm 5     $                | $-1 \pm 10   $                   |
> |  PubMed  | $2.8 \pm 0.3   $                 | $3 \pm 7 $                       |
>
> We will include these results in the revised version of the paper.
>
> With respect to the influence of feature privacy on edge privacy, feature privacy influences how accurately the similarity score between two feature vectors can be computed. In case of perfect feature privacy, the similarity scores are completely random and thus our algorithm would be ineffective in selecting _good_ candidates. In case of no feature privacy, our algorithm would be able to provide a good list of candidate replacement nodes. In both cases, assuming the same number of candidates is available, the same edge-level privacy holds thanks to randomized response. That is, while the two jointly play a role in the utility of the algorithm, their contribution to the privacy of the algorithm is in this case separate. In our experiments we considered the feature privacy budget fixed, as we were interested in exploring, specifically, edge privacy and the role of the edge privacy budget.
>
> **W3**
>
>  We have made the code used for the submission available at this [anonymous repository](https://anonymous.4open.science/r/sp0504/readme.md). We will include it in a revised version of our submission.
>
> **W4**
>
> We briefly address scalability at the end of page 6. For each node we may expect to compute the similarity between $\mathcal{O}(deg(v))^2$ nodes, that is, the pairs of nodes in the two-hop neighborhood. While this may be computationally demanding on a single computing for very dense graphs, we can expect the algorithm to scale well to sparse large datasets (particularly, in efficient practical implementation where the nodes are distributed among several computing units).
>
> **W5**
>
> We will try to make a clearer distinction between the preliminary concepts and our proposed approach.

---

> ### Author Response · Authors · 2024-06-27
> **Reply to Reviewer vXhW - part 2**
>
> With respect to your additional questions:
>
> > In the beginning of Section 4.2, you state [...]
>
> The node-level privacy definition in [1] applies to a _global_ privacy setting. In these settings, some central curator is trusted with the privatization process. In our setting, instead, we focus on a _local_ setting. In fact, Definition 4.1 is provided with respect to the individual nodes in the graph and not with respect to the entire graph (such as Definition 4 in [1], for instance). Finally, we will rephrase our notion of adjacency to make more clear that adjacent neighborhoods should be interpreted in the context of the two-hop extended local view setting we adopt and thus correspond to edge perturbations in the two-hop neighborhood.
>
> > How can the similarity score be adapted in cases where nodes do not have labels? [...]
>
> This is an interesting setting which we have not considered as the benchmark datasets commonly used for private node classification have features. As our neighborhood perturbation preserves the degree of nodes, we can use that information to compute similarities. In case we deemed the node degree to be sensitive information which has to be protected, nodes would exchange a noisy version of the node degree to compute similarities.
>
> > In accordance with W2 [...]
>
> We hope our discussion of W2 helped clarify this aspect. With reference to Figure 2, the (noise free) information of edges, features, and labels is accessed by 3 separate components of our architecture and, after that, only privatized (i.e., noisy) information is used: as we then only post-process the output of these differentially private components, differential privacy is preserved. In this sense, we will try to improve the clarity of the proof in the revised version of the submission.
>
> > How does the algorithm scale with larger datasets? It would be great to provide results on larger datasets (perhaps OGBN-Arxiv?).
>
> While we expect our algorithm to scale well to large sparse graphs, graphs significantly larger than Facebook (such as OGBN-Arxiv, as you suggest), may be difficult to handle with our single GPU setup within the timeframe of the rebuttal. As a choice of datasets, we selected some commonly used ones in the GNN privacy literature with which we could perform comparisons.
>
> > Related to the previous question, how can the model be adapted for datasets that contain edge features? How can this information be used to compute similarities?
>
> In case edge features should be private, one could privatize them similarly to labels (if categorical) or node features (if not). In the simple MPNN settings of a multi-relational graph where the $\text{AGGREGATE}$ function takes edge features into account, we expect one could then utilize the node embeddings to compute similarities with an approach analogous to the one we describe. Note however that our local privacy setting would need to be adapted to determine which entity can have noise free access to the edge features.
>
> > [...] Is there a reason why you did not use the public folds available with some datasets (such as Cora and PubMed)?
>
> As we derive one of our baselines from Sajadmanesh and Gatica-Perez. "Locally private graph neural networks." 2021, who also performed experiments on Cora and PubMed, we decided to adopt an analogous train/val/test split ratio.
>
> > As a suggestion, it would be great to expand your results to the graph classification setting. I believe your algorithm can be easily adapted to this setting as well.
>
> Thank you for the suggestion. The graph classification setting is surely a relevant one for differential privacy. In this case, however, our method would need to take into account multiple graphs in the datasets and the fact that, usually, one considers graphs themselves to be the individual entities to be privatized. While our approach privatizes the edges of the individual graphs, further considerations, research, and comparisons would be necessary to effectively adapt our approach to the graph classification setting. For this reason, we are unable to incorporate the graph classification into the current study. We nevertheless believe this could be a very interesting direction for future work, as differentially private graph classification is an under-explored setting.

---

### Review · Reviewer_zkvH · 2024-06-21

**Summary Of Contributions:**

The authors introduce a framework that utilizes Graph Neural Networks (GNNs) in conjunction with differential privacy techniques to derive representations from graph-structured data while adhering to privacy requirements applicable in real-world scenarios, particularly emphasizing a local setting. To ensure the privacy of features and labels, the graph structure undergoes local privatization through a perturbation strategy, where private nodes are substituted with similar ones based on a similarity score.

**Audience:**

Yes

**Claims And Evidence:**

Yes

**Requested Changes:**

**Motivation:**
- *Privacy Considerations:* Local and global privacy, as well as the limitations of the considered setups in real-world scenarios, should be more thoroughly addressed in the introduction and contributions sections to contextualize the impact of the proposed method for differential privacy (DP). The significance of focusing on the local setting should be elucidated, explaining why this is crucial and whether the proposed method could potentially extend to other settings.

**Algorithm and Experiments:**
- *Baseline Comparisons:* The authors compare their method with only one baseline adapted from two cited methods (Sajadmanesh & Gatica-Perez, 2021, and Wu et al., 2022). It is important to justify why these modifications are necessary and whether other related works could be applied with fewer adjustments.
- *Distinction from Related Work:* Further clarification is needed regarding the distinction from the cited work (Hidano & Murakami, 2022). An experimental comparison should be included, or the authors should justify why such a comparison is not applicable.
- *Similarity Metrics:* Cosine similarity is used as the sole measure for node replacement. The authors should justify this choice and discuss its potential impact on performance. Comparisons with other similarity metrics would provide valuable insights and could serve as a useful ablation study for the proposed algorithm.

**Readability:**
- *Symbol Clarity:* Symbols involved in definitions should be clearly denoted alongside the equations in which they are used. For instance, symbol $m_v$ in Equations (1) and (2) in section 3.1 and $Pr[.]$ in Definition 3.1 should be explicitly defined. Similarly, symbols in algorithms 2, 3, and 4 mentioned in Section 4.2 are hard to identify in the provided text. The basic notations currently provided in the large upper paragraph of page 6 could be repositioned into equations or summarized in a table of basic notations to enhance symbol clarity throughout the paper.

**Strengths And Weaknesses:**

**Strengths:**
- The paper is generally easy to follow and clear in terms of key points in methodology and experiments.
- An experimental evaluation of the proposed method for different datasets and GNN architectures is provided along with a thorough analysis of the results.
- A complete list of hyperparameters for the proposed algorithm is provided in the Appendix, enhancing the reproducibility of the experimental setup.

**Weaknesses:**
- Definition 4.2 as presented is unclear whether it is derived from the cited paper on page 5 (Hidano & Murakami, 2022) or provided by the authors. If it is the latter, the statement that “by construction, node degree is preserved” and the conclusion “while the converse does not hold” require further explanation. Additionally, since the authors claim to provide a definition of ε edge-set LDP for their local privacy settings as a main contribution, it is necessary to clarify any algorithmic or theoretical overlap with the cited works in the methods section, especially concerning edge-set LDP.
- Given that the cited work (Hidano & Murakami, 2022) provides a degree-preserving randomized response, which appears closely related to the proposed method (operating on neighboring lists), comparisons between the two methods are necessary. The authors should justify if such a comparison is not meaningful, by providing more information for the cited work in the related works section.
- For reproducibility purposes (it is not mentioned if the code will be publicly available), the exact hyperparameters of the GNNs need to be specified for the different variants, including the number of layers and hidden dimensions.
- Including a limitations section in the conclusion discussing scenarios where the proposed algorithm may not be applicable would be crucial for guiding future research directions.

---

> ### Author Response · Authors · 2024-06-27
> **Reply to  Reviewer zkvH**
>
> Thank you very much for your review and detailed feedback, which we are happy to address.
>
> > Definition 4.2 as presented [...]
>
> We do not derive our definition from Hidano & Murakami (2022), who proposes an adaptation of randomized response for graph classification, which preserves node degrees in expectation. Instead, we define two adjacent neighborhoods of some node $v$ in terms of sets with the same cardinality, and thus, we do not modify the degree of $v$. Additionally, the algorithm proposed by Hidano & Murakami (2022), while considering local DP, requires that each node knows the total number of nodes in a graph, which is an assumption our local privacy setting does not allow. We will nevertheless improve clarity with respect to the relation between edge set LDP and edge LDP in our revised submission.
>
> > Given that the cited work (Hidano & Murakami, 2022) [...]
>
> Expanding on the response above, Hidano & Murakami (2022) consider a graph classification task and operate on neighbor lists. The use of neighbor lists implies that each node $v$ knows how many (and which) nodes $v$ is not adjacent to. In fact, their approach requires the total number of nodes in the graph to be known by each node. The local privacy setting we adopt, in contrast, does not allow for individual nodes to know the number of nodes of their graph. Moreover, we consider a node classification task where nodes have features and labels while, as mentioned, Hidano & Murakami (2022) consider a graph classification setting for unattributed graphs. We believe that these differences are sufficient to say that a comparison between the two methods is not meaningful. We will include these considerations in the revised version.
>
> > For reproducibility purposes
>
> We have made the code used for the submission available at this [anonymous repository](https://anonymous.4open.science/r/sp0504/readme.md). We will include a link to the repository as well as additional hyperparameters in a revised version of our submission.
>
> > Including a limitations section [...]
>
> Thank you for the observation, we agree that such a section would be beneficial and we will expand our concluding remarks to include a discussion of the limitations of our approach.

---

> ### Author Response · Authors · 2024-06-27
> **Reply to Reviewer zkvH - part 2**
>
> We now address your additional questions.
>
> > Privacy Considerations
>
> We will expand the introduction to provide a better description of local and global differential privacy, and to make it easier to contextualize our contribution. With reference to your previous comment, we will also more explicitly discuss the limitations of our approach and the potential for applications to different settings
>
> > Baseline Comparisons
>
> The modifications we consider are necessary to adapt the baseline we compare against to our local privacy setting, as the original algorithms assume access to the full adjacency matrix, which is not compatible with our privacy setting. We will make these considerations explicit in the submission.
>
> > Distinction from Related Work
>
> Please, refer to the previous comments above which, we hope, sufficiently address this point.
>
> > Similarity Metrics
>
> In preliminary experiments the cosine similarity was a natural choice to compute the similarity between embeddings and provided the most promising results. With respect to potential impact on performance, as the similarity is computed starting from noisy/perturbed feature vectors, it is difficult to estimate how beneficial different metrics may be. Moreover, since our local privacy setting allows no entity to know the features of the entire graphs, a non-parametric/non-learnable notion of similarity may be preferable. We performed some additional experiments using the Euclidean distance to compute similarities, and we report here the results, which show how it delivers a worse privacy-utility trade-off if compared to the cosine similarity, as the cosine similarity obtains better accuracy ($\overline{\Delta_{\text{Acc}}}>0$) and comparable privacy ($\overline{\Delta_{\text{AUC}}}\approx 0 $). We will briefly discuss these considerations in a revised version of the paper as well.
>
> |          | $\overline{\Delta_{\text{Acc}}}$ | $\overline{\Delta_{\text{AUC}}}$ |
> |----------|:------------------------------:|:------------------------------:|
> |   Cora   | $4.8\pm 1.2  $                   | $2 \pm 3 $                       |
> | Facebook | $1.6\pm 0.5  $                   | $1 \pm 1   $                     |
> |  LastFM  | $2.4 \pm 1.4   $                 | $1 \pm 1  $                      |
> |   Photo  | $-0.9 \pm 5 $                    | $-1 \pm 10  $                    |
> |  PubMed  | $2.8 \pm 0.3 $                   | $3 \pm 7 $                       |
>
> > Symbol Clarity
>
> We thank you very much for the suggestion and will work on improving the clarity of our notation.

---

### Review · Reviewer_NPRv · 2024-07-01

**Summary Of Contributions:**

The authors propose GraphPrivatizer that privatizes the structure of a graph under Differential Privacy. It uses controlled perturbation of the graph structure by randomly replacing the neighbors of a node with other similar neighbors, according to some similarity metric. Specifically, authors aggregate features to compute similarities and imposing a minimum similarity score between the original and the replaced nodes provides the best privacy-utility trade-off, and then train a Graph Neural Network server-side without disclosing users’ private information to the server.

**Audience:**

Yes

**Broader Impact Concerns:**

This work investigates a useful topic of research in Differential Privacy for graphs, however I have a concern regarding novelty of the works, outdated baselines (for experiments) and overall architecture (as detailed in the above comments).

**Claims And Evidence:**

Yes

**Requested Changes:**

In addition to the above comments:

Please also provide one figure summarizing where algorithms 2-4 fit in, in terms of the overall architecture (Figure 2).

Please provide dataset properties and statistics summarized in one table (e.g., node count, edge count etc).

The experiment baselines also seem highly outdated, with GCN and GraphSAGE being very old models.

**Strengths And Weaknesses:**

The graph neural network architectures mentioned in the related work are quite outdated, as there are recent state-of-the-art architectures that use bi-level attention based graph aggregation for large-scale multi-relation general-purpose graphs. The authors should mention these works in their related work section for relevance and recency to the latest research:

"Bi-Level Attention Graph Neural Networks," 2021 IEEE International Conference on Data Mining (ICDM), Auckland, New Zealand, 2021, pp. 1126-1131, doi: 10.1109/ICDM51629.2021.00133.

Heterogeneous Graph Attention Network. In The World Wide Web Conference (WWW '19). Association for Computing Machinery, New York, NY, USA, 2022–2032.

Furthermore, the authors should highlight what the novelty of their work is and limitations of existing models, since controlled perturbation of the graph structure by randomly replacing the neighbors of a node with other similar neighbors, according to some similarity metric has been proposed by other prior works.

---

> ### Author Response · Authors · 2024-07-02
> **Reply to Reviewer NPRv**
>
> We thank the reviewer very much for the review and comments, which we are happy to address.
>
> > The graph neural network architectures [...]
>
> In our work, we focused on benchmark datasets commonly used for private learning on graphs that consider a single relation type. A multi-relation setting would require our neighbor perturbation technique to be modified to take into account multiple possible types of relationships. This extension would, however, require some additional considerations with regard to which triples (head, relation, tail) are allowed during the neighborhood perturbation. Considering multi-relational graphs thus is a very interesting possible future direction, which is, however, out of the scope of our current investigation.
>
> With respect to the architectures used, we again considered some of the most commonly used architectures for private learning on graphs, which is what we focus on in the main body of the submission. Nevertheless, we did perform experiments on other, more recent architectures such as, e.g., GATv2, a more recent attention-based architecture. In fact, we strived for a rather model agnostic approach that can be applied to a variety of architectures (while choosing to focus on some of the commonly used ones in related work). These additional results are already available in the original submission in the supplementary materials.
>
> We will in any case include additional considerations in the related work and we thank you for the references.
>
> > Furthermore, the authors should highlight [...]
>
> In our submission we discuss what, to the best of our knowledge, are the most similar approaches to ours and their limitations with respect to the restrictive local privacy setting we consider, and thus the value of our contribution. We are however happy to expand and improve our related work section, as well as to include other work you may point us to which we may not be aware of.
>
> > Please also provide one figure summarizing where algorithms 2-4 fit in, in terms of the overall architecture (Figure 2).
>
> We will redraw figure 2 to make it more evident where algorithms 2-4 act on.
>
> > Please provide dataset properties and statistics summarized in one table (e.g., node count, edge count etc).
>
> This information is available in the appendix (table 4, in the original submission). We report the number of classes, nodes, edges, features, as well as the average node degree for each dataset we used. Is there any additional information we should include?
>
> > The experiment baselines also seem highly outdated, with GCN and GraphSAGE being very old models.
>
> Please refer to our previous comment about the choice of models and, in particular, please note that we performed experiments with more recent architectures as well. We want, moreover, to highlight the fact that our focus is on the perturbation of local neighborhoods, which we strive to perform in a rather model-agnostic way to make our approach applicable to a variety of GNN architectures. In fact, GCN and GraphSAGE are not used themselves as baselines but, rather, are some of the architectures we tested on with respect to some baseline perturbations of the neighborhoods.
>
> > Broader Impact Concerns
>
> We hope that our comments above, which we are of course happy to discuss further, have sufficiently addressed your concerns.

---

> > ### Comment · Reviewer_NPRv · 2024-07-31
> > **Acknowledgement of Review**
> >
> > Dear authors, thank you for providing these useful insights and for addressing my various questions/comments. I have appropriately revised my recommendation for your work.

---

### Author Response · Authors · 2024-07-10
**Revision**

We thank again all the reviewers for their comments and insights.
We have now uploaded a revised version of our manuscript where we addressed your comments.
Specifically, we have:
* expanded our discussion of the related work and of our methodology/approach;
* expanded on our motivation and provided more considerations about local differential privacy;
* increased the clarity of our notation and exposition of section 4.2;
* provided more justifications and experimental results (appendix B) to motivate the choice of the cosine similarity as a similarity metric;
* linked a repository with our code;
* added a discussion of limitations and future work.

We hope these modifications address your comments and we are happy to engage in further discussion to improve our manuscript.

---

### Decision · Action_Editor_Fe4X · 2024-08-17

**Recommendation:** Accept as is

**Comment:**

The paper was reviewed by three expert reviewers. The reviewers initially raised concerns about the presentation of the paper, the employed similarity metric and the scalability of the proposed approach. They also complained about missing related work and lack of discussion on how the proposed approach differs from prior work. Most of these concerns were addressed by the authors in the revision and all reviewers recommended weak acceptance of the paper. I thus think that the paper is now ready for publication. Since the main content of the paper is now longer than 12 pages, the authors should select "Long submission (more than 12 pages of main content)" instead of "Regular submission (no more than 12 pages of main content)" on OpenReview. Furthermore, the authors should add a link to the repository where the code is hosted.

**Audience:**

The problem considered in the paper is gaining prominence within the graph learning community and the paper's findings will be of interest to some individuals in TMLR's audience.

**Claims And Evidence:**

This paper considers a local privacy setting and proposes GraphPrivatizer, a method which provides edge, feature and label privacy. The proposed method seems reasonable. Some theoretical results are also provided in the paper which seem valid. The method's privacy guarantees are empirically evaluated on real-world datasets using privacy attacks.